

# Thermodynamics of adiabatic quantum pumping in quantum dots

Daniele Nello$^\star$ and Alessandro Silva$^\dagger$

International School for Advanced Studies (SISSA), via Bonomea 265, 34136 Trieste, Italy

$\star$ danello@sissa.it , $\dagger$ asilva@sissa.it

## Abstract

We consider adiabatic quantum pumping through a resonant level model, a single-level quantum dot connected to two fermionic leads. Using the tools of adiabatic expansion, we develop a self-contained thermodynamic description of this model accounting for the variation of the energy level of the dot and the tunnelling rates with the thermal baths. This enables us to study various examples of pumping cycles computing the relevant thermodynamic quantities, such as the entropy produced and the dissipated power. These quantities are compared with the transport properties of the system, i.e. the pumped charge and the charge noise. Among other results, we find that the entropy production rate vanishes in the charge quantization limit while the dissipated power is quantized in the same limit.

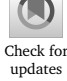
---

## 1  Introduction

The laws of thermodynamics were mainly conceived in the 19th century at the peak of the Industrial Revolution to describe the novel technologies that were being developed in the period, such as steam engines. Classic thermodynamics applies to macroscopic systems while the recent development of nanoscale devices [1] and of quantum information theory posed the problem of reconciling the concepts of thermodynamics with quantum theory, which plays a fundamental role in describing these systems [2,3]. Notable examples of quantum machines are adiabatic quantum pumps: these were first introduced based on the adiabatic theorem by David Thouless in 1983 [4] in the context of isolated quantum systems. A related version of quantum pumping in a transport setting ideal to describe open quantum systems has subsequently been introduced by Piet Brouwer [5], who was able to describe the charge pumped in a cycle pumping through an open, yet non-interacting system, as a geometrical quantity written in terms of the instantaneous scattering matrix of the system (without reference to a specific time dependence). Generalizations of this construction to interacting systems have been attempted [6–8]. For specific settings, characterized by the fact that the conductance is zero along the entire cycle, the charge pumped through a quantum dot can be quantized (with zero associated charge noise [9]), a possibility that makes this physical phenomenon potentially interesting for applications in various areas, such as metrology.

Looking at it as an engine, the operation of a quantum pump should be characterized by standard thermodynamic quantities: the work done, the entropy produced and the heat exchanged. A fresh thermodynamic view of quantum pumping opens up the possibility of addressing qualitatively different questions. For example, is there a minimal work done associated with charge quantization? Or can we find a connection between entropy production and current noise? These issues relating transport to the thermodynamic properties of a pumping cycle can be addressed only by developing a description of transport and thermodynamics within the same formalism (cf. with [10] for classical pumps). In this paper, we focus on this task by addressing adiabatic pumping through the simplest, yet nontrivial system that displays all significant ingredients we are looking for (charge quantization, noise): a resonant level coupled to two leads. We construct our thermodynamic analysis building on the ideas of Bruch et.al. [11] (and other notable works [8, 12–17]) who addressed using the Keldysh technique the thermodynamics of a resonant level whose position is shifted as a function of time. We extend these results to describe quantum pumps and their thermodynamics in terms of a systematic gradient expansion of non-equilibrium Green functions and provide expressions for all relevant thermodynamic quantities characterizing a pumping cycle. Computing thermodynamic quantities for a few examples of cycles we obtain among other things that charge quantization, which is of course attained with zero charge noise, corresponds thermodynamically to zero entropy production (at zero temperature) and a saturated work per cycle proportional to the speed with which the quantity associated to the quantization limit is varied.

The structure of the paper is organized as follows: after introducing the model in Sec. 2 in Sec. 3 we compute transport properties, summarizing the derivation of Brouwer's formula and will introduce a novel way to perform the adiabatic expansion of the noise of the current, showing that it is consistent with the available literature. In Sec. 4.1 and 4.2 we describe the equilibrium thermodynamics of the resonant level model and develop an adiabatic expansion accounting for the variation of both the dot energy level and the level-lead couplings over time. This will allow us to compute the thermodynamic quantities integrated over a cycle in these two parameters and compare them with the results of the transport properties. A thermodynamic tradition is learning by example: we will consider specific thermodynamic cycles in Sec. 5.1-5.2. In these examples, we will compute the relevant quantities averaged over a cycle, enabling us to compare them. This will make it possible to show, among other results, that there is no entropy production (nor noise) for cycles where the charge is quantized. In addition to that, we will show that the dissipated power obeys a similar quantization rule, compared to the one followed by the pumped charge.

## 2 The model

In the following, we will consider adiabatic quantum pumping through a specific system: a time-dependent resonant level model consisting of a single energy level coupled to two metallic leads. The leads act as fermionic reservoirs and are kept fixed at temperature $T$ and chemical potentials $\mu_L$ and $\mu_R$ (from now on assumed to be equal $\mu_L = \mu_R = \mu$).

The Hamiltonian of the system consists of three different terms

$$H = H_D + H_V + H_B \,, \tag{1}$$

where $H_D$ is the Hamiltonian associated with the dot

$$H_D = \epsilon_d(t) d^\dagger d \,. \tag{2}$$

$H_B$ is associated with the leads

$$H_B = \sum_{k\alpha} \epsilon_{k\alpha} c_{k\alpha}^\dagger c_{k\alpha} \,, \tag{3}$$

and $H_V$ to the leads-dot coupling

$$H_V = \sum_\alpha H_V^\alpha = \sum_\alpha V_\alpha(t) \sum_k (d^\dagger c_{k\alpha} + h.c.) \,. \tag{4}$$

Here $d$ is the annihilation operator of the dot level, whilst $c_{k\alpha}$ is associated with an electron with momentum $k$ in the $\alpha = L, R$ lead, and $V_\alpha$ is the coupling between the dot and lead $\alpha$. Throughout this paper we will assume the leads to have a constant density of states and an infinite bandwidth (wideband limit), implying that the decay rate $\Gamma_\alpha = 2\pi \mid V_\alpha \mid^2 \sum_k \delta(\epsilon - \epsilon_{ki})$ does not depend on energy. In this case, the expression for the spectral function of the dot is

$$A(\epsilon) = \frac{\Gamma}{(\epsilon - \epsilon_d)^2 + (\frac{\Gamma}{2})^2} \,, \tag{5}$$

where $\Gamma = \Gamma_L + \Gamma_R$ is the total decay rate.

Adiabatic quantum pumping requires at least two of the system parameters to be varied periodically in time along a certain cycle [5]. We will therefore take both the energy dot level $\epsilon_d(t)$ and the level-dot couplings $V_\alpha(t)$ to be time-dependent and driven by an external agent. Below we will investigate the effect of this external driving on the thermodynamics of the system.

In the following, we will be interested in connecting transport quantities to thermodynamic ones. While the thermodynamics of a quantum pump will be discussed thoroughly below, the study of transport through quantum pumps has been the subject of many studies [5, 13–16, 18–20]. In particular, the two quantities of interest in transport are the charge pumped in a cycle and its noise. Defining the stroboscopic times in terms of the period $T_0$ as $T_n = nT_0$ we may define the operator describing the charge pumped in the n-th period as

$$Q_\alpha^{(n)} = \int_{T_{n-1}}^{T_n} dt\, I_\alpha(t), \tag{6}$$

where $I_\alpha = -dN_\alpha/dt$, with $N_\alpha = \sum_k c_{k,\alpha}^\dagger c_{k,\alpha}$, is the current flowing out of lead $\alpha$. Clearly the average change pumped in M cycles is $Q_\alpha(M) = MQ_\alpha$ where $Q_\alpha = \langle Q_\alpha^{(n)} \rangle$ is the charge pumped in cycle, independent on $n$ in the stationary state.

Coming now to the noise it is evident that current-current correlations produce both fluctuations in the charge pumped in a single cycle as well as correlations of charge pumped in different cycles. The first is described by $\delta Q_\alpha^{(n)} = \langle (Q_\alpha^{(n)})^2 \rangle - (\langle Q_\alpha^{(n)} \rangle)^2$. In the following, however, we will focus on a similar quantity that has the advantage of being similar to the zero frequency component of the noise power spectrum, defined as

$$\delta Q_{\alpha\alpha} = \lim_{M\to+\infty} \frac{(\delta Q_\alpha(M))^2}{M}, \tag{7}$$

where $(\delta Q_{\alpha\alpha}(M))^2 = \sum_{n,m=1}^M (\langle Q_\alpha^{(n)} Q_\alpha^{(m)} \rangle - \langle Q_\alpha^{(n)} \rangle \langle Q_\alpha^{(m)} \rangle)$. Using the definition of the operators we may rewrite the latter as

$$\delta Q_{\alpha\alpha} = \lim_{M\to+\infty} \frac{T_0}{T_M} \int_0^{T_M} dt \int_0^{T_M} dt' [\langle I_\alpha(t) I_\alpha(t') \rangle - \langle I_\alpha(t) \rangle \langle I_\alpha(t') \rangle]. \tag{8}$$

## 3 Pumped charge and its noise

As a warm-up to describe the physical problem we want to address and establish the formalism that will later be used to discuss thermodynamic quantities let us use its most important tool, the gradient expansion, to derive Brouwer's formula for the charge pumped in a cycle by a quantum pump as well as the expressions for its statistical fluctuations. Brouwer's formula will follow the steps reported in Ref. [8]. The current has the following expression

$$\langle I_\alpha \rangle = -\langle \dot{N}_\alpha \rangle = -i\langle [H, N_\alpha] \rangle = i \sum_k (V_\alpha \langle c_{k\alpha}^\dagger d \rangle - h.c). \tag{9}$$

Using standard manipulations with the Keldyish technique (see Appendix A) one can show that the expression of the pumped current is [21]

$$\langle I_\alpha(t) \rangle = \int dt_1 dt_2 \sum_\beta \left[ S_{\alpha\beta}(t, t_1) f(t_1 - t_2) S_{\beta\alpha}^\dagger(t_2, t) - \delta(t - t_1) f(t_1 - t_2) \delta(t_2 - t) \right], \tag{10}$$

where $f(\epsilon) = 1/(\exp[\beta(\epsilon-\mu)]+1)$ is the Fermi distribution function of the leads and $S_{\alpha\beta}(t, t')$ are the time-dependent S-matrices satisfying the unitarity condition

$$\sum_\beta \int dt_1 S_{\alpha\beta}(t, t_1) S_{\alpha\beta}^\dagger(t_1, t') = \delta(t - t'). \tag{11}$$

These expressions can be further simplified in the adiabatic limit, that is when the typical time of variation of system parameters is much longer than the typical time scale of the electron dynamics. For a quantum pump this means to have a period $T_0 \gg 1/\Gamma_{\alpha,\text{av}}$ where $\Gamma_{\alpha,av} = \frac{1}{T_0}\int_0^{T_0} dt\Gamma_\alpha(t)$. In order to perform a gradient expansion [22] on convolutions of the form

$$C(t,t') = \int dt_1 A(t,t_1)B(t_1,t'),\tag{12}$$

we express them in terms of the Wigner coordinates $T = \frac{t+t'}{2}$ and $\tau = t-t'$ and perform the Wigner-Fourier transform with respect to the coordinate $\tau$, defined as

$$A(T,\omega) = \int d\tau e^{i\omega\tau} A(T+\tau/2, T-\tau/2).\tag{13}$$

The Wigner transform of a convolution is not the product of Wigner transforms: instead, we have, formally

$$C(T,\omega) = A(T,\omega)G_{\omega,T}B(T,\omega),\tag{14}$$

where

$$G_{\omega,T} = e^{\frac{1}{2i}(\overleftarrow{\partial_T}\overrightarrow{\partial_\omega} - \overleftarrow{\partial_\omega}\overrightarrow{\partial_T})} = \sum_n \frac{1}{(2i)^n}\frac{1}{n!}(\overleftarrow{\partial_T}\overrightarrow{\partial_\omega} - \overleftarrow{\partial_\omega}\overrightarrow{\partial_T})^n.\tag{15}$$

Expanding the Wigner transform up to first order in the gradients we obtain

$$C(T,\omega) = A(T,\omega)B(T,\omega) + \frac{1}{2i}(\partial_T A(T,\omega)\partial_\omega B(T,\omega) - \partial_\omega A(\omega,T)\partial_T B(T,\omega)).\tag{16}$$

This expansion can be now used to expand systematically Eq. 10 to first order in the gradients. The result is

$$\langle I_\alpha\rangle = \int \frac{d\omega}{2\pi}f(\omega)\Bigg[\sum_\beta\Big\{S_{\alpha\beta}(\omega,T)S_{\beta\alpha}^\dagger(\omega,T) + \frac{1}{2i}(\partial_T S_{\alpha\beta}\partial_\omega S_{\beta\alpha}^\dagger - \partial_\omega S_{\alpha\beta}\partial_T S_{\beta\alpha}^\dagger)\Big\} - 1\Bigg]$$
$$- \sum_\beta\int\frac{d\omega}{4\pi i}(-f'(\omega))\Big[\partial_T S_{\alpha\beta}S_{\beta\alpha}^\dagger - S_{\alpha\beta}\partial_T S_{\beta\alpha}^\dagger\Big].\tag{17}$$

The first term of this sum vanishes due to the gradient expansion of the condition of unitarity of the S-matrix Eq. 11. Therefore we are left only with the last term. Considering the charge pumped in a period $T_0$

$$Q_\alpha = \int_0^{T_0} dt\langle I_\alpha\rangle,\tag{18}$$

and substituting the expression of the current yields

$$Q_\alpha = -\sum_\beta\int\frac{d\omega}{4\pi i}(-f'(\omega))\int_0^{T_0} dT\Big\{\partial_T S_{\alpha\beta}S_{\beta\alpha}^\dagger - S_{\alpha\beta}\partial_T S_{\beta\alpha}^\dagger\Big\}.\tag{19}$$

The dependence on time of the $S$ matrices in the previous expression is to be understood as parametric in the two parameters $(x_1, x_2)$ that define the pumping cycle, i.e. $S_{\alpha\beta}(t) = S_{\alpha\beta}(x_1(t), x_2(t))$. We may therefore use Green's theorem to transform the time integral above, which is just an integral over the pumping cycle $(x_1(t), x_2(t))$, into an integral over the area enclosed by the pumping cycle itself. The result is

$$Q_\alpha = \sum_\beta\int\frac{d\epsilon}{4\pi}f'(\epsilon)\Bigg[\iint_A \frac{dx_1 dx_2}{i}(\partial_{x_2}S_{\alpha\beta}\partial_{x_1}S_{\beta\alpha}^\dagger - \partial_{x_1}S_{\alpha\beta}\partial_{x_2}S_{\beta\alpha}^\dagger)\Bigg].\tag{20}$$

The scattering matrix entering this expression is the instantaneous scattering matrix depending on the varied parameters in time $x_1, x_2$. For the resonant level model, where the time-dependent parameters are the level position and the hybridization strength to the leads, one has, therefore, the following expression ($\alpha, \beta = L, R$)

$$S = \begin{pmatrix} 1 - i\Gamma^L G^R & -i\sqrt{\Gamma_L \Gamma_R} G^R \\ -i\sqrt{\Gamma_L \Gamma_R} G^R & 1 - i\Gamma_R G^R \end{pmatrix}, \tag{21}$$

with

$$G^R = \frac{1}{\epsilon - \epsilon_d + i\frac{\Gamma}{2}}. \tag{22}$$

A similar expansion can be derived to obtain the current noise. The expression we obtain is analogous to those derived in Ref. [21], i.e. the noise can be separated into two different terms

$$\delta Q_{\alpha\alpha} = \frac{T_0}{T_m} \int_0^{T_m} dt\, dt' \int dt_1 dt_2 f(t_1 - t') \tilde{f}(t' - t_2)[\delta(t - t_1)\delta(t - t_2) - S_{\alpha\alpha}^\dagger(t_1, t) S_{\alpha\alpha}(t, t_2)]$$

$$+ \frac{T_0}{T_m} \int_0^{T_m} dt\, dt' \int dt_1 dt_2 f(t' - t_2) \tilde{f}(t_1 - t')[\delta(t - t_1)\delta(t - t_2) - S_{\alpha\alpha}^\dagger(t_1, t) S_{\alpha\alpha}(t, t_2)]$$

$$+ \frac{T_0}{T_m} \int_0^{T_m} dt\, dt' \int dt_1 dt_2 dt_1' dt_2' f(t_1 - t_2') \tilde{f}(t_1' - t_2)$$

$$* \sum_{\gamma\delta} [S_{\alpha\gamma}^\dagger(t_1, t) S_{\alpha\delta}(t, t_2) S_{\delta\alpha}^\dagger(t_1', t') S_{\gamma\alpha}(t', t_2') - \delta(t - t_1)\delta(t' - t_1')\delta(t - t_2)\delta(t' - t_2')], \tag{23}$$

where $\tilde{f}(t, t') = \delta(t - t') - f(t, t')$.

To perform the adiabatic expansion of both terms, we notice that they have the same structure as a product of convolutions. Taking $m \to +\infty$ and performing an expansion in the gradients as done before for the current one obtains at zero order

$$\delta Q_{\alpha\alpha}^{(0)} = -2 \int \frac{d\epsilon}{2\pi} \left( -\frac{1}{\beta} \frac{\partial f}{\partial \epsilon} \right) \int_0^{T_0} dT \left( 1 - S_{\alpha\alpha}(\epsilon, T) S_{\alpha\alpha}^\dagger(\epsilon, T) \right). \tag{24}$$

where $\beta$ is the inverse temperature. This is the average over a period of the instantaneous equilibrium Johnson-Nyquist noise [23].

At first order in the gradients the only non-zero contribution is

$$\delta Q_{\alpha\alpha}^{(1),th} = \int_0^{T_0} dT \int \frac{d\epsilon}{4\pi i} \left( -\frac{1}{\beta} \frac{\partial^2 f}{\partial \epsilon^2} \right) \sum_{\beta \neq \alpha} \left[ \partial_T S_{\alpha\beta} S_{\alpha\beta}^\dagger - S_{\alpha\beta} \partial_T S_{\alpha\beta}^\dagger \right]$$

$$- \int_0^{T_0} dT \int \frac{d\epsilon}{4\pi i} \left( -\frac{1}{\beta} \frac{\partial f}{\partial \epsilon} \right) \sum_\beta \left[ \partial_\epsilon S_{\alpha\beta} \partial_T S_{\alpha\beta}^\dagger - \partial_T S_{\alpha\beta} \partial_\epsilon S_{\alpha\beta}^\dagger \right]. \tag{25}$$

This term, which obviously depends on the operation of the pump, is a first-order contribution to thermal noise proportional to the temperature and vanishing at zero temperature [18, 24].

The gradient expansion performed above turns out to miss an important shot noise term and is valid only when $\hbar\Omega \ll k_B T$, where $\Omega = \frac{2\pi}{T_0}$. The finite shot-noise contribution which survives even at zero order was first computed in Ref. [18]. It arises from the emission/absorption of quanta of energy from the scatterer. The expression of this zero-temperature shot noise is

$$\delta Q_{\alpha\alpha}^{(1),sh} = \sum_{q=1}^\infty \frac{q}{4\pi} C_{\alpha\alpha,q}^{(sym)}(0), \tag{26}$$

where

$$C_{\alpha\alpha,q}^{(sym)}(E) = \frac{C_{\alpha\alpha,q}(E) + C_{\alpha\alpha,-q}(-E)}{2}, \tag{27}$$

$$C_{\alpha\alpha,q}(E) = \sum_{\gamma\delta} [S_{\alpha\gamma}^*(E)S_{\alpha\delta}(E)]_q [S_{\alpha\delta}^*(E)S_{\alpha\gamma}(E)]_{-q}, \tag{28}$$

which arises from the quartic term of eq. 23. The superscript $[\ ]_q$ identifies the Fourier coefficients, defined as

$$[A]_q(E) = \int_0^{T_0} \frac{dt}{T_0} e^{iq\Omega t} A(E,t). \tag{29}$$

The derivation of the present shot noise term is described in Appendix E. Moreover, the relevance of the various terms of the noise is discussed in more detail in the Appendix F. In Sec. 5 we will analyze the noise obtained together with thermodynamic quantities to gain further insight into the relationship between transport and thermodynamics.

## 4 Quantum pump as an engine

Now that the transport problem is described in its full generality, let us look at a quantum pump as a thermodynamic engine. By varying the parameters $(x_1, x_2)$ over a cycle it is clear that we are performing a certain work on the system as well as dissipating heat and generating entropy. The goal of this section will be to give concrete expressions to these quantities for the specific problem of quantum pumping of a resonant level model. Of course, as in the case of transport properties, we will have to proceed in steps, first considering the quasi-static limit, and then proceeding to higher-order contributions in the gradient expansion.

### 4.1 Quasistatic limit

Let us start developing this formalism in the limit of reversible and quasi-static transformations where we can work in the equilibrium gran-canonical framework at fixed temperature $\beta^{-1}$ and chemical potential $\mu$. Evaluating the grand potential of the total system, $\Omega = -1/\beta \ln \Xi$, where $\Xi = Tr[e^{-\beta(H-\mu N)}]$, in terms of the density of states $\rho(\epsilon)$ of the total system one obtains

$$\Omega_{tot} = -\frac{1}{\beta} \int \frac{d\epsilon}{2\pi} \rho(\epsilon) \ln[1 + e^{-\beta(\epsilon-\mu)}]. \tag{30}$$

We are now interested in extracting the time-dependent part of this expression when parameters are varied quasi-statically: for a resonant level model in which the time-dependent parameters are $\epsilon_d(t), \Gamma_{L/R}(t)$ this amounts, as shown in Appendix B, to the replacement in Eq.(30) of the total density of states $\rho(\epsilon)$ with the *instantaneous* local spectral function of the dot $A_t(\omega) = A(\omega, [\epsilon_d(t), \Gamma_{L/R}(t)]) = -2\text{Im}[G^r(\omega, [\epsilon_d(t), \Gamma_{L/R}(t)])]$ obtaining an *instantaneous* grand potential

$$\Omega_t = -\frac{1}{\beta} \int \frac{d\epsilon}{2\pi} A_t(\epsilon) \ln[1 + e^{-\beta(\epsilon-\mu)}]. \tag{31}$$

From this expression, we can derive the quasistatic thermodynamic functions $N_t^{(0)}$, $S_t^{(0)}$ and $E_t^{(0)}$, respectively particle number, entropy, energy [11] obtaining

$$N_t^{(0)} = \int \frac{d\epsilon}{2\pi} A_t(\epsilon) f(\epsilon), \tag{32}$$

$$S_t^{(0)} = k_B \int \frac{d\epsilon}{2\pi} A_t(\epsilon) \Big[ -f \ln f - (1-f) \ln(1-f) \Big], \tag{33}$$

$$E_t^{(0)} = \int \frac{d\epsilon}{2\pi} \epsilon A_t(\epsilon) f(\epsilon). \tag{34}$$

Clearly, the derivatives of these quantities with respect to time are connected to the reversible energy change $\dot{E}^{(1)}$, the reversible power $\dot{W}^{(1)}$, the heat exchange rate $\dot{Q}^{(1)}$ and the current $\dot{N}^{(1)}$. In particular, using the relation $\partial_\Gamma A = -\partial_\epsilon Re(G^R)$ the expression for the reversible power $\dot{W}^{(1)} = \dot{\epsilon}_d \partial_{\epsilon_d} \Omega + \sum_\alpha \dot{\Gamma}_\alpha \partial_{\Gamma_i} \Omega$ can be written as

$$\dot{W}^{(1)} = \dot{\epsilon}_d \int \frac{d\epsilon}{2\pi} A f + \dot{\Gamma} \int \frac{d\epsilon}{2\pi} Re(G^R) f. \tag{35}$$

Similar calculations lead to the expression of the quasi-static heat exchange rate as

$$\dot{Q}^{(1)} = T \frac{dS^{(0)}}{dt} = \dot{\epsilon}_d \int \frac{d\epsilon}{2\pi} (\epsilon - \mu) A \partial_\epsilon f + \dot{\Gamma} \int \frac{d\epsilon}{2\pi} (\epsilon - \mu) Re(G^R) \partial_\epsilon f. \tag{36}$$

The current out of the dot is

$$\dot{N}^{(1)} = \frac{dN^{(0)}}{dt} = \dot{\epsilon}_d \int \frac{d\epsilon}{2\pi} A \partial_\epsilon f + \dot{\Gamma} \int \frac{d\epsilon}{2\pi} Re(G^R) \partial_\epsilon f. \tag{37}$$

Finally, the energy exchange rate

$$\dot{E}^{(1)} = \frac{dE^{(0)}}{dt} = -\dot{\epsilon}_d \int \frac{d\epsilon}{2\pi} \epsilon \partial_\epsilon A f - \dot{\Gamma} \int \frac{d\epsilon}{2\pi} \epsilon f \partial_\epsilon Re(G^R). \tag{38}$$

Notice that these quantities satisfy the first law of thermodynamics in the form

$$\dot{E}^{(1)} = \dot{W}^{(1)} + \dot{Q}^{(1)} + \mu \dot{N}^{(1)}. \tag{39}$$

## 4.2 Gradient expansion of thermodynamic quantities

Let us now come to the main results of this paper: a self-contained thermodynamic description of the operation of a quantum pump. Quantum pumping is not a quasi-static phenomenon: the quasi-static contribution to the pumped current is just zero. It is intuitively appealing that the same will be true for certain thermodynamic quantities that are expected to be intimately connected to the flow of a current, such as the heat dissipated and the entropy produced. Therefore in order to address them we will have to extend our analysis to account for corrections to the quasi-static limit using a gradient expansion. Our goal will be for each generic quantity to express it as expansion in gradients as $\mathcal{O} = \sum_i \mathcal{O}^{(i)}$, where $\mathcal{O}^{(i)}$ contains the i-th time derivative. In order to do so we will first write $\mathcal{O}$ in terms of non-equilibrium Green's functions (Appendix C) and then perform their adiabatic expansion deriving the next-order corrections to the expressions obtained in the previous section. The expansion we are going to derive is an expansion in gradients precisely as the one obtained for the pumped charge and the noise, i.e. using as small parameters $\dot{\epsilon}_d/\Gamma^2$ and $\dot{\Gamma}_\alpha/\Gamma^2$.

Let us start with the simplest quantity: the particle number in the resonant level. The average number of particles is readily connected to a Green's function using its definition, $N = \langle d^\dagger d \rangle = -iG^<(t,t)$. Therefore one can identify $N^{(i)}$ with the i-th order expansion in the gradients of the lesser Green function (see Appendix C) which can be calculated starting from the Keldysh equation

$$G^< = \int dt_1 dt_2 G^R(t,t_1) \Sigma^<(t_1,t_2) G^A(t_2,t').$$ (40)

Performing a gradient expansion of this one readily obtains the zeroth order terms reported above and

$$N^{(1)} = -\frac{\dot{\epsilon}_d}{2} \int \frac{d\epsilon}{2\pi} \partial_\epsilon f A^2 - \frac{\dot{\Gamma}}{2} \int \frac{d\epsilon}{2\pi} \partial_\epsilon f \frac{A^2}{\Gamma}(\epsilon - \epsilon_d).$$ (41)

This result can be used to compute the second-order correction to the current out of the dot

$$\begin{aligned}
\dot{N}^{(2)} = &-\frac{\dot{\epsilon}_d^2}{2} \int \frac{d\epsilon}{2\pi} \partial_\epsilon^2 f A^2 - \frac{\dot{\Gamma}^2}{2} \int \frac{d\epsilon}{2\pi} \partial_\epsilon f (\epsilon - \epsilon_d) \partial_\Gamma \left(\frac{A^2}{\Gamma}\right) \\
&-\frac{\dot{\epsilon}_d \dot{\Gamma}}{2} \int \frac{d\epsilon}{2\pi} \left[\partial_\Gamma A^2 - \frac{A^2}{\Gamma} + \frac{\partial_\epsilon A^2}{\Gamma}(\epsilon - \epsilon_d)\right] \partial_\epsilon f \\
&-\frac{\ddot{\epsilon}_d}{2} \int \frac{d\epsilon}{2\pi} \partial_\epsilon f A^2 - \frac{\ddot{\Gamma}}{2} \int \frac{d\epsilon}{2\pi} \partial_\epsilon f \frac{A^2}{\Gamma}(\epsilon - \epsilon_d).
\end{aligned}$$ (42)

The argument becomes more involved if one wants to calculate the gradient expansion of the entropy. For this sake one needs to substitute in the expression for the entropy introduced the Fermi distribution $f$ with the non-equilibrium distribution $\phi(\epsilon, T)$ [12] obtained from the Wigner transform of the lesser Green's function $G^<(\epsilon, T) = iA(\epsilon, T)\phi(\epsilon, T)$

$$S = k_B \int \frac{d\epsilon}{2\pi} A \left[ -\phi \ln \phi - (1 - \phi) \ln(1 - \phi) \right].$$ (43)

A gradient expansion of the lesser Green's function (and a similar one for the retarded one) results in a gradient expansion for the non-equilibrium distribution, hence for the entropy. The results for $\phi$ are given in Appendix C. The resulting expansion to the first order of the non-equilibrium distribution gives $S^{(1)}$

$$S^{(1)} = -\frac{k_B \dot{\epsilon}_d}{2} \int \frac{d\epsilon}{2\pi} \left(\frac{\epsilon - \mu}{k_B T}\right) \partial_\epsilon f A^2 - \frac{k_B \dot{\Gamma}}{2} \int \frac{d\epsilon}{2\pi} \left(\frac{\epsilon - \mu}{k_B T}\right) \partial_\epsilon f \frac{A^2}{\Gamma}(\epsilon - \epsilon_d),$$ (44)

and therefore the entropy production rate to the second order is

$$\begin{aligned}
\dot{S}^{(2)} = &\frac{\dot{\epsilon}_d^2}{2T} \int \frac{d\epsilon}{2\pi}(\epsilon - \mu)\partial_\epsilon f \partial_\epsilon A^2 - \frac{\dot{\Gamma}^2}{2T} \int \frac{d\epsilon}{2\pi}(\epsilon - \mu)\partial_\epsilon f (\epsilon - \epsilon_d)\partial_\Gamma\left(\frac{A^2}{\Gamma}\right) \\
&-\frac{\dot{\epsilon}_d \dot{\Gamma}}{2T} \int \frac{d\epsilon}{2\pi}(\epsilon - \mu)\partial_\epsilon f \left[\partial_\Gamma A^2 - \frac{\partial_\epsilon A^2}{\Gamma}(\epsilon - \epsilon_d) + \frac{A^2}{\Gamma}\right] - \frac{\ddot{\epsilon}_d}{2T} \int \frac{d\epsilon}{2\pi}(\epsilon - \mu)\partial_\epsilon f A^2 \\
&-\frac{\ddot{\Gamma}}{2T} \int \frac{d\epsilon}{2\pi}(\epsilon - \mu)\partial_\epsilon f \frac{A^2}{\Gamma}(\epsilon - \epsilon_d).
\end{aligned}$$ (45)

Coming to the internal energy, it can be verified that at zero order $E^{(0)} = \langle H_D \rangle^{(0)} + \frac{1}{2}\langle H_V \rangle^{(0)}$ [12]. We may then identify an "effective system" of Hamiltonian $H_{eff} = H_D + \frac{1}{2}H_V$ and an "effective bath" $H_B + \frac{1}{2}H_V$ [11]. Therefore, at every order in a gradient expansion, we have

$$E^{(i)} = \langle H_D \rangle^{(i)} + \frac{1}{2}\langle H_V \rangle^{(i)}.$$ (46)

The expectation value for $\langle H_V \rangle$ is given in Appendix D. As for the internal energy we compute the zeroth order in the gradients (quasi-static approximation) in Eq.(34) and the rate of change to first order in Eq.(38). Let us now evaluate the first-order correction of the energy

$$E^{(1)} = -\frac{\dot{\epsilon}_d}{2} \int \frac{d\epsilon}{2\pi} \epsilon \partial_\epsilon f A^2 - \frac{\dot{\Gamma}}{2} \int \frac{d\epsilon}{2\pi} \epsilon \partial_\epsilon f \frac{A^2}{\Gamma}(\epsilon - \epsilon_d),$$ (47)

whose time derivative reads

$$\dot{E}^{(2)} = \frac{\dot{\epsilon}_d^2}{2} \int \frac{d\epsilon}{2\pi} \epsilon \partial_\epsilon f \, \partial_\epsilon A^2 - \frac{\dot{\Gamma}^2}{2} \int \frac{d\epsilon}{2\pi} \epsilon \partial_\epsilon f (\epsilon - \epsilon_d) \partial_\Gamma \left(\frac{A^2}{\Gamma}\right)$$
$$+ \frac{\dot{\epsilon}_d \dot{\Gamma}}{2} \int \frac{d\epsilon}{2\pi} \left[ \epsilon \partial_\epsilon f \, \partial_\Gamma A^2 + \epsilon \frac{\partial_\epsilon A^2}{\Gamma} \partial_\epsilon f (\epsilon - \epsilon_d) \right] - \frac{\ddot{\epsilon}_d}{2} \int \frac{d\epsilon}{2\pi} \epsilon \partial_\epsilon f A^2$$
$$- \frac{\ddot{\Gamma}}{2} \int \frac{d\epsilon}{2\pi} \epsilon \partial_\epsilon f \frac{A^2}{\Gamma}(\epsilon - \epsilon_d).$$ (48)

The expression for the energy agrees with the energy-resolved one [12]

$$E^{(i)} = \int \frac{d\epsilon}{2\pi} \epsilon A(\epsilon, T) \phi^{(i)}(\epsilon, T).$$ (49)

The power can be computed according to the definition. We can distinguish two different contributions, relative to $H_D$ and $H_V$. The first one is

$$\dot{W}_D^{(i)} = \left\langle \frac{\partial H_D}{\partial \epsilon_d} \right\rangle^{(i-1)} \dot{\epsilon}_d = \dot{\epsilon}_d N^{(i-1)}.$$ (50)

Likewise for the components of the coupling

$$\dot{W}_V^{(i)} = \sum_\alpha \left\langle \frac{\partial H_V}{\partial V_\alpha} \right\rangle^{(i-1)} \dot{V}_\alpha = \sum_\alpha \dot{V}_\alpha(t) \sum_k (\langle d^\dagger c_{k\alpha} \rangle^{(i)} + h.c.) = \sum_\alpha \frac{\dot{V}_\alpha(t)}{V_\alpha(t)} \langle H_V^\alpha \rangle^{(i)}.$$ (51)

Changing variable to $\Gamma_\alpha$

$$\dot{W}_V^{(i)} = \sum_\alpha \frac{\dot{\Gamma}_\alpha}{2\Gamma_\alpha} \langle H_V^\alpha \rangle^{(i-1)},$$ (52)

so that

$$\dot{W}^{(i)} = \dot{W}_V^{(i)} + \dot{W}_D^{(i)}.$$ (53)

The first order in the gradients of the power was computed in Eq. 35. We may now use the expansion above to compute the second order correction as

$$\dot{W}^{(2)} = -\frac{\dot{\epsilon}_d^2}{2} \int \frac{d\epsilon}{2\pi} \partial_\epsilon f A^2 - \frac{\dot{\Gamma}^2}{4} \int \frac{d\epsilon}{2\pi} \partial_\epsilon f \, \partial_\Gamma A + \frac{\dot{\epsilon}_d \dot{\Gamma}}{2} \int \frac{d\epsilon}{2\pi} \partial_\epsilon f \, \partial_\epsilon A + \sum_\alpha \frac{\dot{\Gamma}_\alpha^2}{2\Gamma_\alpha} \int \frac{d\epsilon}{2\pi} f \frac{\partial_\epsilon A}{2}.$$ (54)

Note the presence of $\Gamma_\alpha$ at the denominator: this term causes a singularity at $\Gamma_\alpha = 0$, which appears only when multiple heat baths are present.

The heat exchange rate $\mathcal{Q}$ cannot be calculated directly [25] since there are no physical process accounting for dissipation and the Landauer-like picture of transport assumes that dissipation processes take place far away from the system and do not affect its dynamics [26]. The only way it can be derived is from the first law of thermodynamics. The latter reads

$$\dot{E}^{(i)} = \dot{W}^{(i)} + \dot{\mathcal{Q}}^{(i)} + \mu \dot{N}^{(i)}.$$ (55)

Then, the heat exchange has to be calculated inverting this relation

$$\dot{Q}^{(i)} = \dot{E}^{(i)} - \dot{W}^{(i)} - \mu \dot{N}^{(i)}. \tag{56}$$

Finally, the heat exchange flow reads

$$
\begin{aligned}
\dot{Q}^{(2)} = &-\frac{\dot{\epsilon}_d^2}{2}\int\frac{d\epsilon}{2\pi}(\epsilon-\mu)\partial_\epsilon^2 f A^2 - \frac{\dot{\Gamma}^2}{2}\int\frac{d\epsilon}{2\pi}\partial_\epsilon f(\epsilon-\epsilon_d)(\epsilon_d-\mu)\partial_\Gamma\left(\frac{A^2}{\Gamma}\right) \\
&-\frac{\dot{\Gamma}^2}{4}\int\frac{d\epsilon}{2\pi}\partial_\epsilon^2 f(\epsilon-\epsilon_d)\partial_\Gamma A - \sum_\alpha\frac{\dot{\Gamma}_\alpha^2}{2\Gamma_\alpha}\int\frac{d\epsilon}{2\pi}f\frac{\partial_\epsilon A}{2} \\
&-\frac{\ddot{\epsilon}_d}{2}\int\frac{d\epsilon}{2\pi}(\epsilon-\mu)\partial_\epsilon f A^2 - \frac{\ddot{\Gamma}}{2}\int\frac{d\epsilon}{2\pi}(\epsilon-\mu)\partial_\epsilon f\frac{A^2}{\Gamma}(\epsilon-\epsilon_d) \\
&-\frac{\dot{\epsilon}_d\dot{\Gamma}}{2}\int\frac{d\epsilon}{2\pi}\partial_\epsilon f(\epsilon_d-\mu)\left[\partial_\Gamma A^2 - \frac{A^2}{\Gamma} + \frac{\partial_\epsilon A^2}{\Gamma}(\epsilon-\epsilon_d)\right] \\
&+\frac{\dot{\epsilon}_d\dot{\Gamma}}{2}\int\frac{d\epsilon}{2\pi}\partial_\epsilon f(\epsilon-\epsilon_d)\partial_\Gamma A^2 - \frac{\dot{\epsilon}_d\dot{\Gamma}}{4}\int\frac{d\epsilon}{2\pi}\partial_\epsilon^2 f(\epsilon-\epsilon_d)\partial_\epsilon A. \tag{57}
\end{aligned}
$$

The results obtained for $N$ and $\dot{W}$ are consistent with the ones found in [27].

A final comment on the adiabatic expansion we have just performed is that the entropy production rate is not simply related only to the heat production, but there is a further contribution which can be identified with the dissipated power. In particular this relation holds

$$\dot{S}^{(2)} = \frac{\dot{Q}^{(2)}}{T} + \frac{\dot{W}^{(2)}}{T}. \tag{58}$$

This is because in general, we have this expression

$$\frac{dS}{dt} = \dot{\Sigma} + \frac{\dot{Q}}{T}, \tag{59}$$

where $\dot{\Sigma}$ can be related to the entropy production of the universe and $S$ is the entropy of the system. In turn, the change of the entropy of the universe is given by the mismatch between the corresponding reversible work rate and the work rate in an irreversible process, called the unusable energy, i.e.

$$T\dot{\Sigma} = \dot{E}_{un} = \dot{W}_{rev} - \dot{W}, \tag{60}$$

which up to second order corresponds to $\dot{W}^{(2)}$ [28].

# 5 Quantum thermodynamics of various pumping cycles

It is now time to put all the pieces of the puzzle together and show how the formulas developed above can be used to gain insight into the physics of quantum pumping by combining information on thermodynamic quantities as well as transport properties (pumped charge and its noise). We will do so for various examples of cycles constructed for a resonant-level model.

## 5.1 The peristaltic cycle

The simplest cycle one can think of is the "peristaltic" cycle which consists of four strokes as shown in Fig.1: the level initially empty at $+\epsilon_0$ coupled only to $\Gamma_L$. It is then lowered to $-\epsilon_0$ and filled with an electron from the left. Afterwards, the coupling is switched to the right and the level is raised again to $+\epsilon_0$ and emptied on the right. Finally, the coupling is reinstated

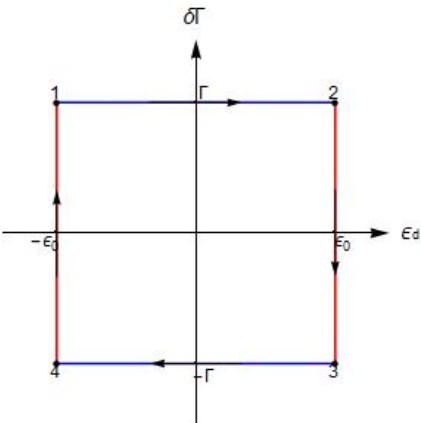

Figure 1: The peristaltic cycle: the level initially empty at $+\epsilon_0$ coupled only to $\Gamma_L$. It is then lowered to $-\epsilon_0$ and filled with an electron from the left. Afterwards, the coupling is switched to the right and the level is raised again to $+\epsilon_0$ and emptied on the right. Finally, the coupling is reinstated to its initial value.

to its initial value. In this cycle, the only varied quantities are $\epsilon_d$ and $\delta\Gamma = \Gamma_L - \Gamma_R$. The total decay rate $\Gamma = \Gamma_L + \Gamma_R$ is not changed.

We calculate the current, using Brouwer's formula (Sec. 3). In the limit of low temperature $T$, the pumped charge is

$$Q_L^{(0)} = \frac{2}{\pi}\left\{\arctan(x) + \frac{x}{1+x^2}\right\} + O(T^2),\tag{61}$$

where $x = \frac{\epsilon_0}{\Gamma/2}$. In the limit $x \gg 1$, the pumped charge is quantized, $Q_L^{(0)} \to 1$.

The expectation is that the corresponding noise will tend to zero in the quantization limit. We are interested in the zero-temperature limit, which enjoys contributions only from the "shot" noise (Sec. 3). The zeroth order contribution to the current noise is just thermal and vanishes in this limit

$$\delta Q_{LL}^{(0)} = 0.\tag{62}$$

We evaluate the first-order term in the gradients, containing the relevant shot noise contribution. The result of this calculation gives a current noise which tends to zero in the quantization limit

$$\lim_{x\to\infty} \delta Q_{LL}(x) = 0\tag{63}$$

(see e.g. Fig 2).

Let us now proceed with the comparison with thermodynamic quantities by integrating the previously calculated ones over a cycle. Clearly, integrating over a cycle quantities expanded up to the first order will give a quantity independent of the cycle parametrization, hence geometric. In the present case, however, the integrals are all vanishing, as one can see from the one of the power expanded to first order

$$\dot{W}^{(1)} = \frac{\partial\Omega}{\partial\epsilon_d}\dot{\epsilon}_d + \frac{\partial\Omega}{\partial\Gamma}\dot{\Gamma},\tag{64}$$

which using Green's theorem can be expressed as

$$W_{cycle}^{(0)} = \int_0^{T_0} dt\,\dot{W}^{(1)} = \iint_A d\epsilon_d d\Gamma\left[-\frac{\partial^2\Omega}{\partial\epsilon_d\partial\Gamma} + \frac{\partial^2\Omega}{\partial\Gamma\partial\epsilon_d}\right] = 0.\tag{65}$$

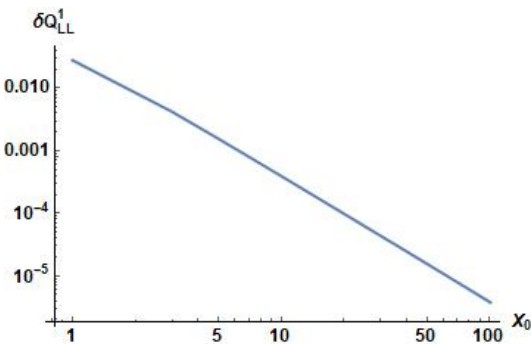

Figure 2: The first order of the noise integrated over the peristaltic cycle in the zero-temperature limit (in logarithmic scale). In this case, we have set $v_{\epsilon_d} = |\dot{\epsilon}_d|$ and $v_{\delta\Gamma} = |\dot{\delta\Gamma}|$ to 1. Since the sum of Eq. 26 cannot be carried out to infinity numerically, we chose a value of $q_{max}$ as an upper limit to that sum. The latter value was chosen to guarantee convergence and was set to $q_{max} = 10000$.

Similar reasoning can be applied for integrals of all other rates at first order. We can point out that this is a trivial consequence of the absence of any chemical potential or temperature difference.

In contrast, second-order rates integrated over a cycle will not be geometric quantities and will depend on the specific parameterization. In the following, we will consider a linear parameterization so that the rates $\dot{\epsilon}_d$ and $\dot{\delta\Gamma}$ are constants along the four strokes. The work would display some divergent integrals, signalling the failure of the gradient expansion for this particular cycle (see Appendix G). The problem concerns indeed the fact that the gradient expansion can be justified only if both $\Gamma_L, \Gamma_R \neq 0$ at all times, which is of course inconsistent with the condition $\delta\Gamma = \pm\Gamma$. One way out is to regularize the cycle, making $\delta\Gamma$ vary from $[-\Gamma + \delta, \Gamma - \delta]$. While the pumped charge and the other terms in the current noise turn out to have a smooth limit as $\delta \to 0$, the work should be computed only for $\delta$ small but finite. The work per cycle is

$$W_{cycle}^{(1)} = -v_{\epsilon_d} \int_{-\epsilon_0}^{\epsilon_0} d\epsilon_d \int \frac{d\epsilon}{2\pi} \partial_\epsilon f A^2 + \frac{v_{\delta\Gamma}}{2} \int_{-\Gamma+\delta}^{\Gamma-\delta} d\delta\Gamma \frac{\Gamma}{\Gamma^2 - \delta\Gamma^2} \int \frac{d\epsilon}{2\pi} f \frac{\partial_\epsilon A}{2}. \tag{66}$$

Performing the integration over $\epsilon_d$ one obtains in the limit $T \to 0$

$$W_{cycle}^{(1)} = v_{\epsilon_d} \Delta W_{\epsilon_d}^{(1)} + v_{\delta\Gamma} \Delta W_{\delta\Gamma}^{(1)}, \tag{67}$$

where $v_{\epsilon_d} = |\dot{\epsilon}_d|$, $v_{\delta\Gamma} = |\dot{\delta\Gamma}|$, and the coefficients are

$$\Delta W_{\epsilon_d}^{(1)} = \frac{2}{\pi\Gamma} \left\{ \arctan x + \frac{x}{1+x^2} \right\}, \tag{68}$$

and

$$\Delta W_{\delta\Gamma}^{(1)} = \frac{1}{\Gamma} \operatorname{arctanh} \left[ 1 - \xi \right] \frac{1}{\pi} \frac{1}{x^2 + 1}, \tag{69}$$

in terms of the dimensionless variables $x = \frac{\epsilon_0}{\Gamma/2}$ and $\xi = \frac{\delta}{\Gamma}$. Comparing this result with the one obtained for the charge pumped over a cycle, we obtain the following direct relationship $\Delta W_{\epsilon_d}^{(1)} = \frac{v_{\epsilon_d}}{\Gamma} Q_L^{(0)}$. The contribution proportional to $v_{\delta\Gamma}$ is shown in Fig. 4 and it is vanishing in the limit $x \to \infty$. Therefore in the limit of charge quantization

$$W_{cycle}^{(1)} = \frac{v_{\epsilon_d}}{\Gamma}, \tag{70}$$

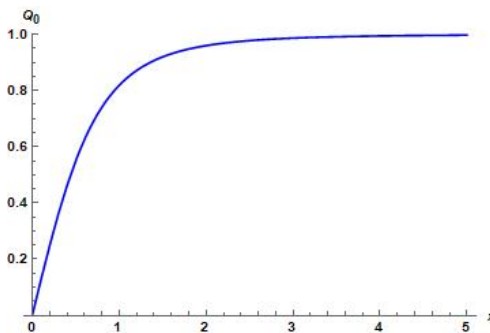

Figure 3: The current integrated over a cycle for the left lead w.r.t. the adimensional parameter x. It is clear that in the limit $x \to \infty$ the quantity displays a quantization to 1.

the work is finite in the presence of charge quantization and proportional to the pumped charge, in the case of constant velocity.

Let us now compute in the same way the entropy produced over a cycle

$$S_{cycle}^{(1)} = \frac{v_{\epsilon_d}}{T} \int_{-\epsilon_0}^{\epsilon_0} d\epsilon_d \int \frac{d\epsilon}{2\pi} (\epsilon - \mu) \partial_\epsilon f \, \partial_\epsilon A^2 \,. \tag{71}$$

Performing the integrations the result up to the first order in the temperature is

$$S_{cycle}^{(1)} = v_{\epsilon_d} \Delta S_{\epsilon_d}^{(1)} \,, \tag{72}$$

where

$$\Delta S_{\epsilon_d}^{(1)} = \frac{\pi T k_B^2}{3} \frac{128}{\Gamma^3} \frac{x}{(1+x^2)^3} + O(T^3) \,. \tag{73}$$

Comparing this expression with the expression of the work, we observe that while in the limit $x \to \infty$ the work saturates to the value $\frac{v_{\epsilon_d}}{\Gamma}$, the entropy production tends to zero (Fig. 5).

For completeness, we report here also the result for the remaining quantities

$$\mathcal{Q}_{cycle}^{(1)} = -v_{\epsilon_d} \int_{-\epsilon_0}^{\epsilon_0} d\epsilon_d \int \frac{d\epsilon}{2\pi} (\epsilon - \mu) \partial_\epsilon^2 f A^2 - \frac{v_{\delta\Gamma}}{2} \int_{-\Gamma+\delta}^{\Gamma-\delta} d\delta\Gamma \frac{\Gamma}{\Gamma^2 - \delta\Gamma^2} \int \frac{d\epsilon}{2\pi} f \frac{\partial_\epsilon A}{2} \,, \tag{74}$$

$$E_{cycle}^{(1)} = v_{\epsilon_d} \int_{-\epsilon_0}^{\epsilon_0} d\epsilon_d \int \frac{d\epsilon}{2\pi} \epsilon \partial_\epsilon f \, \partial_\epsilon A^2 \,, \tag{75}$$

$$N_{cycle}^{(1)} = -v_{\epsilon_d} \int_{-\epsilon_0}^{\epsilon_0} d\epsilon_d \int \frac{d\epsilon}{2\pi} \partial_\epsilon^2 f A^2 \,. \tag{76}$$

Performing the integrations one obtains that in the limit $x \to \infty$ and $T \to 0$ $N_{cycle}^{(1)} = 0$, $S_{cycle}^{(1)} = 0$ and $\mathcal{Q}_{cycle}^{(1)} = -W_{cycle}^{(1)}$. These conditions define a Non-Equilibrium Steady State (NESS) (see for example [29]).

## 5.2 Other examples of cycles

In our analysis of the peristaltic cycle we found that in the limit of quantization, the work per cycle saturates to a value determined by the rate of change of the energy level $v_{\epsilon_d}$. In contrast, the entropy produced per cycle goes to zero (as the noise). The purpose of this section will be to explore other examples of a cycle and study the possibility that these qualitative results could apply to more general situations.

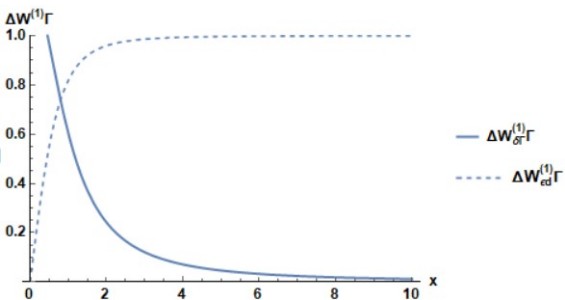

Figure 4: $\Delta W^{(1)}_{\epsilon_d}\Gamma$ and $\Delta W^{(1)}_{\delta\Gamma}\Gamma$. The contribution proportional to $v_{\epsilon_d}$ tends to 1 for $x \to \infty$, following the direct relation with the pumped charge. In contrast, the contribution proportional to $v_{\delta\Gamma}$ tends to zero in the same limit.

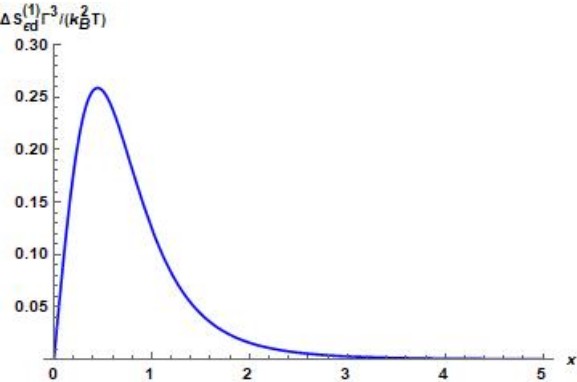

Figure 5: The leading order in temperature of the entropy vs. the adimensional parameter x. From this figure, it is evident that the quantity tends to zero in the quantization limit.

Let us consider a modification of the peristaltic cycle in which the couplings are modified one by one and not together: the level is lowered from $\epsilon_d = \epsilon_0$ to $-\epsilon_0$, while connected mainly to the right lead with $\Gamma_R = \Gamma_0$ and $\Gamma_L = \delta$ ($\xi = \delta/\Gamma_0 \ll 1$). The role of the two couplings is then inverted by first raising $\Gamma_L$ to $\Gamma_0$ and then decreasing $\Gamma_R$ to $\delta$. It is now the turn of the level to be raised from $-\epsilon_0$ to $\epsilon_0$. Then the $\Gamma_R$ and $\Gamma_L$ are exchanged again with the inverse of the above process. The results for the pumped charge are portrayed in Fig. 6. It displays indeed charge quantization in the large $x = \epsilon_0/\Gamma$ limit. The current noise can be safely computed for this cycle and the results for the two coefficients which govern the dependence on $v_{\epsilon_d}$ and $v_\Gamma$ are in Fig. 7. As we can see both the coefficients tend to $\xi = \frac{\delta}{\Gamma}$ as $x \to \infty$. Therefore $\lim_{\xi \to 0} \lim_{x \to \infty} \delta Q_{\alpha\alpha}(x, \xi) = 0$.

Let us now turn to the work per cycle which has the following expression

$$W^{(1)}_{cycle} = v_{\epsilon d}\Delta W^{(1)}_{\epsilon_d} + v_\Gamma \Delta W^{(1)}_\Gamma, \tag{77}$$

where

$$\Delta W^{(1)}_{\epsilon_d} = \frac{2}{\Gamma_0 \pi}\left\{\arctan(x) + \frac{x}{1+x^2}\right\}, \tag{78}$$

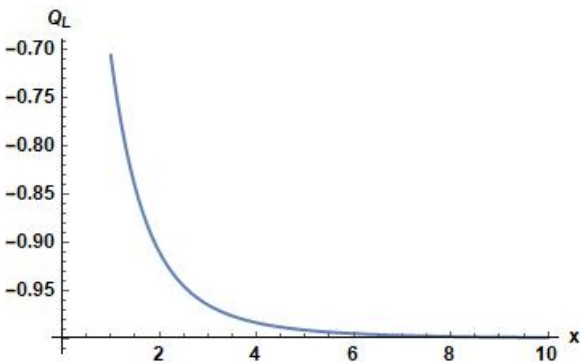

Figure 6: The charge pumped from left to right. It is negative because the current flows from left to right. The current displays charge quantization in the large x limit.

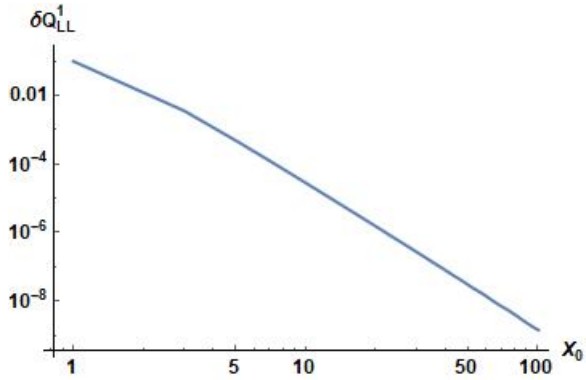

Figure 7: The first order of the noise of the current $\delta Q_{LL}$ (in logarithmic scale).The current noise tends to 0 as $x \to \infty$ as expected, following the quantization requirement.

and

$$\Delta W_{\Gamma}^{(1)} = \frac{1}{\pi\Gamma_0} \frac{1}{(x^2+1)} \left\{ -2x \operatorname{arccot}\left[\frac{x}{\xi-2}\right] \right.$$
$$-2x \operatorname{arccot}\left[\frac{x}{\xi+2}\right] + 2\log\left[\frac{1-\xi}{\xi}\right] + \log\left[\frac{\xi^2+x^2+2\xi x+1}{\xi^2+x^2-2\xi x+4}\right] \right\}$$
$$+ \frac{2}{\pi\Gamma_0}\left(\frac{2-\xi}{x^2+(2-\xi)^2} - \frac{1+\xi}{x^2+(1+\xi)^2}\right), \tag{79}$$

displays a quantization of $\frac{W_{cycle}^{(1)}\Gamma_0}{v_{\epsilon_d}} \to 1$ in the large x limit (see Fig. 8). This appears to be in agreement with what is stated for the peristaltic cycle.

Finally, let us focus on the entropy whose first non-zero order reads

$$S_{cycle}^{(1)} = v_\Gamma \Delta S_\Gamma^{(1)} + v_{\epsilon d} \Delta S_{\epsilon_d}^{(1)} + O(T^3), \tag{80}$$

where

$$\Delta S_\Gamma^{(1)} = \frac{\pi}{12}\frac{1}{\Gamma^3}(k_B^2 T)\left[\frac{2}{(4x^2+1)^2} - \frac{16}{(x^2+1)^2}\right], \tag{81}$$

and

$$\Delta S_{\epsilon_d}^{(1)} = \frac{\pi}{3}(k_B^2 T)\frac{1}{\Gamma^3}\frac{32x}{(1+x^2)^3}. \tag{82}$$

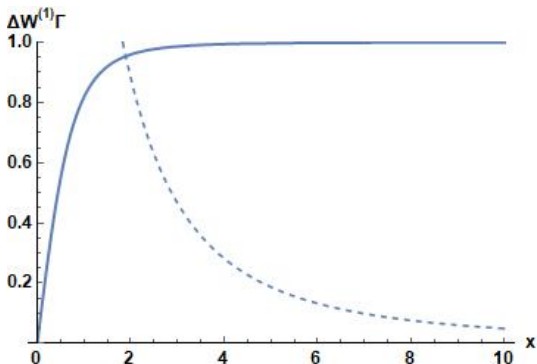

Figure 8: The two components of the dissipated power proportional to $\nu_{\epsilon_d}$ and $\nu_\Gamma$ for $\xi = 0.001$. The work displays a quantization to $\frac{\nu_{\epsilon_d}}{\Gamma_0}$ in the limit $x \to \infty$, as the first component is saturated to 1, while the second vanishes in this limit.

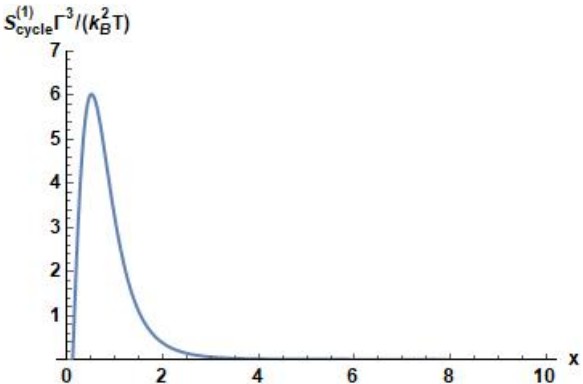

Figure 9: The second order in temperature of the entropy for $\nu_\Gamma = \nu_{\epsilon_d} = 1$. We see that both the coefficients vanish in the quantization limit.

The two coefficients are in Fig. 9. We can see that also in this case, the entropy tends to zero in the charge quantization limit. All the other quantities do not add any relevant physical information: as before since $S_{cycle}^{(1)} = 0$, $\mathcal{Q}_{cycle}^{(1)} = -W_{cycle}^{(1)}$ all the external work is converted into dissipated heat.

Our previous example shows that the main results pertaining to the peristaltic cycle, i.e. work quantization in the limit of quantized charge, no entropy production and zero noise, pertain also to similar cycles.

Let us now focus on another cycle whose peculiarity is to have a maximal pumped charge equal to half an electron charge: the triangular cycle introduced in [30]. In this cycle, at the beginning, the dot is weakly coupled with strength $\delta$ to both leads. Then, it is loaded by coupling to the left lead up to $\Gamma_0 \gg \delta$. The next step is to shift the coupling from the left to the right reservoir. Finally, the dot is discharged while returning to the initial state. The energy level of the dot is maintained constant $\epsilon_d = \epsilon_0$ and only the couplings are varied ( see Fig. 10). In this example, we have half an electron per period. This means that the current noise has to be finite. The fact that the charge transport is on average equal to $1/2$ per cycle, means that half of the times one charge is transported and the other half none. The charge pumped per cycle is plotted in Fig. 11 and is given by the expression

$$Q_R^{(1)} = \frac{2}{\pi} \int_{I(C)} dX_L dX_R \frac{X}{[1+X^2]^2} = \frac{1}{\pi} \left[ \arctan[X_0] - \frac{X_0}{1+X_0^2} \right], \tag{83}$$

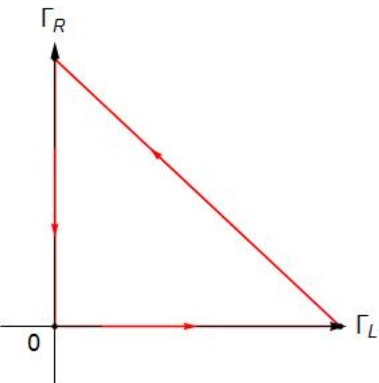

Figure 10: The diagram of the cycle with fractional charge quantization.

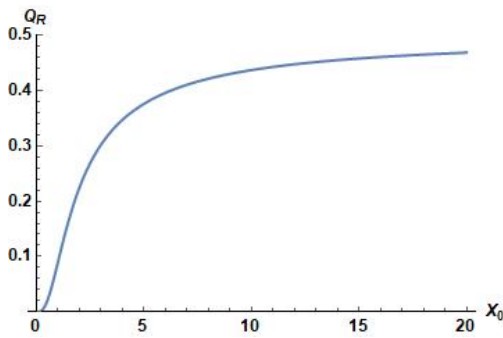

Figure 11: The charge pumped w.r.t the adimensional parameter $X_0$, displaying the fractional quantization to the value $\frac{1}{2}$.

where $I(C)$ is the triangular contour in Fig. 10. In terms of $X = \frac{\Gamma}{2\epsilon_0}$ and $X_0 = \frac{\Gamma_0}{2\epsilon_0}$. It is evident that

$$\lim_{X_0 \to \infty} Q_R^{(1)} = 1/2 \,. \tag{84}$$

The adiabatic current noise in Fig. 12 is finite in the quantization limit, due to the non-integer charge. In this case, the most relevant contribution is from the hypotenuse of the process, where both the couplings are varied and the energy level is half occupied on average. The value reached by the charge noise is

$$\lim_{X_0 \to \infty} \delta Q_{RR}^{(1)} = \frac{1}{2}\left(1 - \frac{1}{2}\right) = \frac{1}{4} \,. \tag{85}$$

The work done per cycle (Fig. 13) is

$$W_{cycle}^{(1)} = v_\Gamma \Delta W_\Gamma^{(1)} + v_{\delta\Gamma} \Delta W_{\delta\Gamma}^{(1)} \,, \tag{86}$$

where

$$\Delta W_\Gamma^{(1)} = \frac{1}{\epsilon_0 \pi}\left[\frac{X_0}{1 + X_0^2} + \arctan[X_0] - \frac{\delta}{1 + \delta^2} - \arctan[\delta]\right] \,, \tag{87}$$

and

$$\Delta W_{\delta\Gamma}^{(1)} = \frac{1}{\Gamma} \operatorname{arctanh}\left[1 - \delta\right] \frac{1}{\pi} \frac{2X_0}{X_0^2 + 1} \,, \tag{88}$$

with $\delta = \frac{\eta}{\Gamma_0}$. We have, in the quantization limit

$$\frac{W_{cycle}^{(1)} \epsilon_0}{v_\Gamma} \to \frac{1}{2} \,. \tag{89}$$

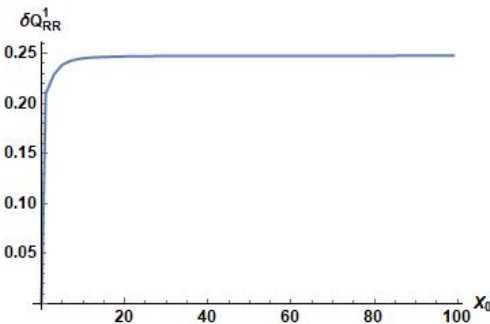

Figure 12: The noise with respect to $X_0$. In this case, we observe a finite noise in the quantization limit. The limiting value is $\frac{1}{4}$.

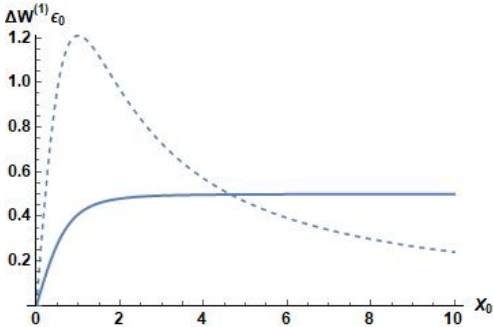

Figure 13: The two components of the work with respect to $X_0$. The figure shows how in the limit $X_0 \to \infty$ the work tends to the value $W_{cycle}^{(1)} \to \frac{1}{2}\frac{v_\Gamma}{\epsilon_0}$.

The entropy integrated over a cycle (fig. 14) reads

$$S_{cycle}^{(1)} = v_\Gamma \Delta S_\Gamma^{(1)}, \tag{90}$$

$$\Delta S_\Gamma^{(1)} = \frac{1}{\epsilon_0^3}\frac{\pi}{12}k_B^2 T\frac{2X_0}{(1+X_0^2)^2}. \tag{91}$$

Likewise, the entropy in the quantization limit tends to zero.

The results for the two processes we have considered up to now indicate that, when the design of the cycle allows us to define a limiting condition which entails a quantized charge pumped, there are some conclusions we can draw regarding the other quantities relevant to our purposes. In particular, the out-of-equilibrium work performed on the system obeys a similar quantization relation. The entropy integrated over the cycle vanishes in the considered limit. Likewise, the charge noise disappears, as the charge pumped is quantized.

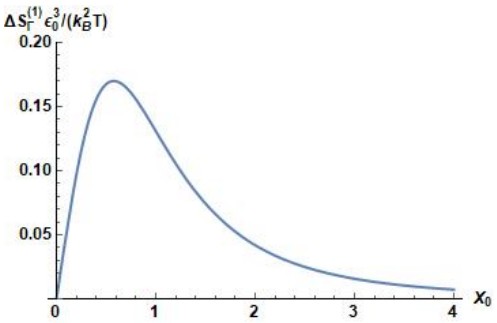

Figure 14: The entropy with respect to $X_0$.

## 6  Conclusion and outlook

In this paper, we have studied the problem of characterizing thermodynamically adiabatic pumping through a single energy-level quantum dot coupled to two leads. We established a systematic scheme based on a gradient expansion to calculate all the relevant transport and thermodynamic quantities.

For a resonant level model, focusing on specific cycles and comparing transport to thermodynamic quantities we have shown that whenever a quantized charge is attained one expects, together with zero charge noise, zero entropy production and a saturated work per cycle proportional to the speed with which the quantity associated to the quantization limit is varied.

The methods developed here could in principle be generalized to other time-dependent transport problems, such as transport through multilevel dots or even interacting systems [7, 31]. Another interesting direction of future related research is to study quantum stochastic thermodynamic quantities (for example the work distribution) and how fluctuation theorems can apply to these thermodynamic cycles.

The results we obtained are relative to a specific model, but nonetheless offer insight into the phenomenon of adiabatic pumping and its thermodynamical implications, which can be relevant in other contexts (for example Thouless pumps).

## Acknowledgments

We thank Yuval Gefen for numerous discussions on this subject and his important insight on the topic.

**Funding information** AS gratefully acknowledges support from PNRR MUR project PE0000023-NQSTI and the Quantera project "SuperLink".

## A  Derivation of Eq. 10

Starting from the expression of the current and using the lesser Green function $G_{d,k\alpha}^{<}(t',t) = i\langle c_{k\alpha}^{\dagger}(t)d(t')\rangle$ we can write

$$\langle I_\alpha \rangle = 2Re\left\{\sum_k V_\alpha G_{d,k\alpha}^{<}(t,t)\right\}. \tag{A.1}$$

It is straightforward to see that, whenever the leads are non-interacting, the lesser Green's function entering the definition of the current has the expression [22]

$$G_{d,k\alpha}^{<}(t,t') = \int dt_1 \left[ g_{k\alpha}^{<}(t,t_1)[V_\alpha]^* G^A(t_1,t') + g_{k\alpha}^{r}(t,t_1)[V_\alpha]^* G^{<}(t_1,t') \right]. \tag{A.2}$$

Performing now the summation over $k$ as in Eq. A.1 using $g_{k,\alpha}^{<}(\epsilon) = 2\pi i \delta(\epsilon - \epsilon_k) f(\epsilon)$ and $g_{k,\alpha}^{r}(\epsilon) = 1/(\epsilon - \epsilon_k + i0^+)$, where $f(\epsilon) = 1/(\exp[\beta(\epsilon - \mu)] + 1)$ is the Fermi function, we have

$$\sum_k G_{k\alpha}^{<}(t,t') = \int dt_1 \left[ 2\pi \nu_0 i f(t-t_1)[V_\alpha]^* G^A(t_1,t') - \pi \nu_0 i [V_\alpha]^* G^{<}(t_1,t') \right], \tag{A.3}$$

where $\nu_0 = \sum_k \delta(\epsilon - \epsilon_{k\alpha})$ is the constant density of states and $f(t)$ is the (properly regularized) Fourier transform of the Fermi distribution. Therefore the expectation value of the current

operator can be written as

$$\langle I_\alpha \rangle = 2Re\left[i\int dt_1 f(t-t_1)T_{\alpha\alpha}^a(t_1,t') - \frac{i}{2}T_{\alpha\alpha}^<(t,t)\right], \tag{A.4}$$

in terms of the generalized, time-dependent transmission matrices

$$T_{\alpha\beta}^{R,A,\gtrless}(t,t') = 2\pi\nu_0\sum_k [V_\alpha(t)]^* G^{R,A,\gtrless}(t,t')V_\beta(t'). \tag{A.5}$$

A further simplification of Eq. A.4 is obtained by expressing of $T_{\alpha\alpha}^<$ in terms of retarded and advanced quantities using

$$G^<(t,t') = \int dt_1 dt_2 G^R(t,t_1)\Sigma^<(t_1,t_2)G^A(t_2,t), \tag{A.6}$$

with

$$\Sigma^<(t,t') = 2\pi\nu_0 i\sum_\alpha V_\alpha(t)f(t-t')[V_\alpha(t')]^*. \tag{A.7}$$

Then we have

$$T_{\alpha\alpha}^<(t,t) = \sum_\beta\int dt_1 dt_2 T_{\alpha\beta}^R(t,t_1)f(t_1-t_2)T_{\beta\alpha}^A(t_2,t). \tag{A.8}$$

Substituting this expression into the current we obtain

$$\langle I_\alpha \rangle = i\int dt_1[f(t-t_1)T_{\alpha\alpha}^R(t_1,t) - T_{\alpha\alpha}^A(t,t_1)f(t_1-t)]$$
$$+ \sum_\beta\int dt_1 dt_2 T_{\alpha\beta}^R(t,t_1)f(t_1-t_2)T_{\beta\alpha}^A(t_2,t). \tag{A.9}$$

Finally introducing the time-dependent scattering matrices defined as

$$S_{\alpha\beta} = \delta_{\alpha\beta}\delta(t-t') + iT_{\alpha\beta}^R(t,t'), \tag{}$$

we arrive at Eq. 10.

## B The density of states

The density of states is given as the trace of the spectral function of the system over all the single-particle states $n$

$$\rho(\epsilon) = \sum_n A_{nn}(\epsilon), \tag{B.1}$$

and we remind that $A_{nn} = -2Im G_{nn}^R$ in terms of the retarded Green function. In the basis of uncoupled dot and leads electron states we can decompose the density of states in terms of the dot and leads contributions

$$\rho(\epsilon) = A_{dd}(\epsilon) + \sum_{k\alpha} A_{kk,\alpha}(\epsilon). \tag{B.2}$$

To calculate $A_{kk,\alpha}$ we write the Dyson equation for the retarded Green function of the leads

$$G_{kk,\alpha}^R(\epsilon) = g_{k\alpha}^R(\epsilon) + (g_{k\alpha}^R(\epsilon))^2 V_\alpha^2(t)G_{dd}^R(\epsilon). \tag{B.3}$$

Since $\Sigma^R_{k\alpha} = |V_\alpha(t)|^2 g^R_{k\alpha}(\epsilon)$, where $g^R_{k\alpha}(\epsilon)$ is the free lead green function, we define $\sigma_\alpha(\epsilon) = \sum_k \Sigma^R_{k\alpha}$ one can rewrite the above density of states in terms of the retarded self-energy as

$$\rho(\epsilon) = A_{dd}(\epsilon)\left(1 - \frac{d}{d\epsilon}Re(\sigma_\alpha(\epsilon))\right) + 2Re(G^R_{dd}(\epsilon))\frac{d}{d\epsilon}Im(\sigma_\alpha(\epsilon)) + \nu_\alpha(\epsilon), \quad \text{(B.4)}$$

where $\nu_\alpha(\epsilon) = -2\sum_k Im(g^R_{k\alpha}(\epsilon))$.

In our case, it follows from the definition that

$$\sigma_\alpha = -\frac{i}{2}\Gamma_\alpha. \quad \text{(B.5)}$$

As a consequence, the terms that depend on the derivatives of the self-energy vanish. Furthermore, the free density of states of the leads depends neither on $\epsilon_d$ nor on $\Gamma_\alpha$ so that we can cast it aside. What we are left with is simply the density of states of the dot alone $A_{dd}$, which we indicate as $A$ in the course of the article.

## C  The expansion of the equations of motion

In the following, we derive the gradient expansion of different Green functions.

- We start with the gradient expansion of the equation of motion for the retarded Green function of the dot

$$\delta(t - t') = \int dt_1 G^R(t, t_1)[i\partial_{t_1}\delta(t_1 - t') - \epsilon_d(t_1)\delta(t_1 - t') - \Sigma^R(t_1 - t')], \quad \text{(C.1)}$$

with the retarded self-energy $\Sigma^R(t, t') = \sum_{k\alpha} |V_\alpha(t)|^2 g^R_{k\alpha}(t, t')$ We switch to the Wigner transform, defined as

$$G(\epsilon, t) = \int d\tau G(t_1, t_2)e^{i\epsilon\tau}, \quad \text{(C.2)}$$

where $t = \frac{t_1 + t_2}{2}$ and $\tau = t_1 - t_2$. We recall that for a convolution, the Wigner transform is

$$\int dt' C(t_1, t')D(t', t_2) = \int \frac{d\epsilon}{2\pi}e^{-i\epsilon\tau}C(\epsilon, \tau) * D(\epsilon, \tau), \quad \text{(C.3)}$$

where $C(\epsilon, \tau) * D(\epsilon, \tau) = C(\epsilon, \tau)Exp[\frac{i}{2}(\overleftarrow{\partial_\epsilon}\overrightarrow{\partial_t} - \overleftarrow{\partial_t}\overrightarrow{\partial_\epsilon})]D(\epsilon, \tau)$. Therefore, up to the first order we have

$$1 = G^R(\epsilon, \tau)\left[\epsilon - \epsilon_d + \frac{i}{2}\Gamma\right] + \frac{i}{2}\left[\partial_\epsilon G^R(\epsilon, \tau)(-\dot{\epsilon}_d(t) + \frac{i}{2}\dot{\Gamma}) - \partial_t G^R(\epsilon, t)\right]. \quad \text{(C.4)}$$

Therefore, up to the first order in the velocities, for the retarded and advanced Green functions we have $G^R(\epsilon, t) = (\epsilon - \epsilon_d(t) + i\frac{\Gamma(t)}{2})^{-1}$ and $G^A(\epsilon, t) = (\epsilon - \epsilon_d(t) - i\frac{\Gamma(t)}{2})^{-1}$.

- The lesser component of the Green function is given by

$$G^< = \int dt_1 dt_2 G^R(t, t_1)\Sigma^<(t_1, t_2)G^A(t_2, t'). \quad \text{(C.5)}$$

The zero order is

$$G^{<(0)}(\epsilon, t) = G^R \Sigma^< G^A = iAf. \quad \text{(C.6)}$$

As we already know, the part dependent on $\epsilon_d$ yields a contribution $-i\frac{\dot{\epsilon}_d}{2}\partial_\epsilon f A^2$. Now, let's work out the different contributions to the part which is dependent on $\dot{\Gamma}$

1)
$$\frac{i}{2}\left(\partial_\epsilon G^R \partial_t \Sigma^< G^A - G^R \partial_t \Sigma^< \partial_\epsilon G^A\right) = \frac{i}{2}(i\dot{\Gamma}f)(\partial_\epsilon G^R G^A - G^R \partial_\epsilon G^A)$$
$$= -\frac{i}{2}\dot{\Gamma}f\frac{A^2}{\Gamma}, \qquad (C.7)$$

2)
$$\frac{i}{2}\left(-\partial_t G^R \partial_\epsilon \Sigma^< G^A + G^R \partial_\epsilon \Sigma^< \partial_t G^A\right) = \frac{i}{2}\dot{\Gamma}(i\Gamma\partial_\epsilon f)\frac{i}{2}([G^R]^2 G^A + G^R[G^A]^2)$$
$$= -\frac{i}{2}\dot{\Gamma}\partial_\epsilon f\frac{A^2}{\Gamma}(\epsilon - \epsilon_d(t)), \qquad (C.8)$$

3)
$$\frac{i}{2}\left(\partial_\epsilon G^R \Sigma^< \partial_t G^A - \partial_t G^R \Sigma^< \partial_\epsilon G^A\right) = \frac{i}{2}\dot{\Gamma}(i\Gamma f)\frac{i}{2}\left(\partial_\epsilon G^R[G^A]^2 + [G^R]^2\partial_\epsilon G^A\right)$$
$$= \frac{i}{2}\dot{\Gamma}f\frac{A^2}{\Gamma}, \qquad (C.9)$$

where we used the following relations: $\partial_\epsilon G^R G^A - G^R \partial_\epsilon G^A = i\frac{A^2}{\Gamma}$, $\partial_t G^{R/A} = -\dot{\epsilon}_d \partial_\epsilon G^{R/A} + \dot{\Gamma}(\mp\frac{i}{2})[G^{R/A}]^2$, $Re(G^R) = \frac{\epsilon - \epsilon_d}{\Gamma}A$ and $[G^R]^2[G^A]^2 = (\frac{A}{\Gamma})^2$. Overall, we obtain

$$G^<(\epsilon, t) = iAf - i\frac{\dot{\epsilon}_d}{2}\partial_\epsilon f A^2 - i\frac{\dot{\Gamma}}{2}\partial_\epsilon f\frac{A^2}{\Gamma}(\epsilon - \epsilon_d). \qquad (C.10)$$

As a consequence, we can identify a non-equilibrium distribution function

$$\phi = f - \frac{\dot{\epsilon}_d}{2}\partial_\epsilon f A - \frac{\dot{\Gamma}}{2}\partial_\epsilon f\frac{A}{\Gamma}(\epsilon - \epsilon_d). \qquad (C.11)$$

# D  Calculation of the expectation value of the coupling term

Now, let us derive the expression of $\langle H_V \rangle$ up to the first order from the expansion of the mixed Green function. To calculate $\langle H_V \rangle$ we write

$$\langle H_V \rangle = \sum_\alpha V_\alpha(t) \sum_k \left[\langle d^\dagger c_{k\alpha}\rangle + \langle c_{k\alpha}^\dagger d\rangle\right] = \sum_\alpha \langle H_V^\alpha \rangle. \qquad (D.1)$$

Where we define $H_V^\alpha = V_\alpha(t)\sum_k\left[d^\dagger c_{k\alpha} + c_{k\alpha}^\dagger d\right]$. It reads

$$\langle H_V^\alpha \rangle = 2V_\alpha(t)\sum_k Im\left[G_{d,k\alpha}^<(t,t)\right], \qquad (D.2)$$

with $G_{d,k\alpha}^< = i\langle c_{k\alpha}^\dagger(t')d(t)\rangle$, for which the property $G_{d,k\alpha}^<(t,t) = -\left(G_{k\alpha,d}^<(t,t)\right)^*$ holds. Now, the equation of motion for the mixed Green function leads to

$$\langle H_V^\alpha \rangle = 2V_\alpha(t)\sum_k Im\left(\int dt'[G^R(t,t')g_{k\alpha}^<(t',t) + G^<(t,t')g_{k\alpha}^A(t',t)]\right)$$
$$= 2Im\left(\int dt'[G^R(t,t')\Sigma_\alpha^<(t',t) + G^<(t,t')\Sigma_\alpha^A(t',t)]\right). \qquad (D.3)$$

Moving to the Wigner transform

$$\langle H_V^\alpha \rangle = 2Im\left(\int \frac{d\epsilon}{2\pi}[G^R(\epsilon,t)*\Sigma_\alpha^<(\epsilon,t) + G^<(\epsilon,t)*\Sigma_\alpha^A(\epsilon,t)]\right). \qquad (D.4)$$

The second term $G^<(\epsilon,t)*\Sigma_\alpha^A(\epsilon,t)=G^<(\epsilon,t)*(\frac{i}{2}\Gamma_\alpha)$ dose not contribute. In fact, the zero-order term is real and to the next order we can apply this type of argument

$$\int\frac{d\epsilon}{2\pi}Im\left(\frac{i}{2}\partial_\epsilon G^<(\epsilon,t)\frac{i}{2}\dot{\Gamma}_\alpha\right)=\frac{\dot{\Gamma}_\alpha}{4}\int\frac{d\epsilon}{2\pi}\frac{\partial_\epsilon G^<(\epsilon,t)}{i}=\frac{\dot{\Gamma}_\alpha}{8\pi}\left[\frac{\partial_\epsilon G^<(\epsilon,t)}{i}\right]_{-\infty}^{+\infty}=0. \quad (D.5)$$

Up to the first order in the velocity, the gradient expansion yields

$$\langle H_V^\alpha\rangle=2Im\left(\int\frac{d\epsilon}{2\pi}\left[G^R(\epsilon,t)if(\epsilon)\Gamma_\alpha-\frac{i}{2}\partial_t G^R(\epsilon,t)i\partial_\epsilon f\Gamma_\alpha+\frac{i}{2}\partial_\epsilon G^R(\epsilon,t)if(\epsilon)\dot{\Gamma}_\alpha\right]\right). \quad (D.6)$$

The gradient expansion of the whole interaction term therefore reads:

$$\langle H_V\rangle=2Im\left(\int\frac{d\epsilon}{2\pi}\left[G^R(\epsilon,t)if(\epsilon)\Gamma-\frac{i}{2}\partial_t G^R(\epsilon,t)i\partial_\epsilon f\Gamma+\frac{i}{2}\partial_\epsilon G^R(\epsilon,t)if(\epsilon)\dot{\Gamma}\right]\right). \quad (D.7)$$

# E Derivation of the shot noise term in the first order of the adiabatic expansion

To derive the "shot" noise term in the first order of the adiabatic expansion of the current fluctuations, which is not present in the gradient expansion of the latter quantity, we link up with the approach employed in [18], namely the adiabatic expansion of the Floquet scattering matrix. Furthermore, we show that this approach yields the same results for the adiabatic expansion of all the other thermodynamics and transport quantities we have considered in the article.

The definition of the two-times scattering matrix relates the outgoing states to the ongoing ones

$$b_\alpha(t)=\sum_\beta\int_{-\infty}^\infty dt_1 S_{\alpha\beta}(t,t_1)a_\beta(t_1). \quad (E.1)$$

If we perform the Fourier transform of this expression, what we obtain is

$$b_\alpha(\epsilon)=\sum_\beta\int_{-\infty}^\infty\frac{d\omega}{2\pi}S_{\alpha\beta}(\epsilon,\epsilon+\omega)a_\beta(\epsilon+\omega), \quad (E.2)$$

adopting the ingoing energy $\epsilon$ as a reference. But we have to note that the scattering matrix elements are periodic in their arguments $S(t,t')=S(t,t'+T_0)$. So, it's more appropriate to use a Fourier series expansion

$$b_\alpha(\epsilon)=\sum_\beta\sum_{n=-\infty}^\infty S_{\alpha\beta}^F(\epsilon,\epsilon_n)a_\beta(\epsilon_n), \quad (E.3)$$

where $\epsilon_n=\epsilon+n\Omega$, and $\Omega=\frac{2\pi}{T_0}$. The matrix $S^F(\epsilon,\epsilon_n)$ is dubbed "Floquet scattering matrix" and described in a series of articles, most prominently [13].

The definition of the current noise is

$$\delta I_{\alpha\alpha}(t,t')=\langle\Delta I_\alpha(t)\Delta I_\alpha(t')\rangle, \quad (E.4)$$

and $\Delta I_\alpha(t)=I_\alpha(t)-\langle I_\alpha(t)\rangle$. It can be rewritten as

$$\delta I_{\alpha\alpha}(t,t')=\langle I_\alpha(t)I_\alpha(t')\rangle-\langle I_\alpha(t)\rangle\langle I_\alpha(t')\rangle. \quad (E.5)$$

The current operator, in turn, reads

$$I_\alpha(t) = b_\alpha^\dagger(t)b_\alpha(t) - a_\alpha^\dagger(t)a_\alpha(t). \tag{E.6}$$

In the Fourier transform, the noise of the current turns into

$$\delta I_{\alpha\alpha}(t,t') = \int \frac{dE\,dE'\,dE''\,dE'''}{(2\pi)^4} e^{i(E-E')t} e^{i(E''-E''')t'} \tag{E.7}$$

$$\times \Bigg[ \langle (b_\alpha^\dagger(E)b_\alpha(E') - a_\alpha^\dagger(E)a_\alpha(E'))(b_\alpha^\dagger(E'')b_\alpha(E''') - a_\alpha^\dagger(E'')a_\alpha(E''')) \rangle$$

$$- (\langle b_\alpha^\dagger(E)b_\alpha(E')\rangle - \langle a_\alpha^\dagger(E)a_\alpha(E')\rangle)(\langle b_\alpha^\dagger(E'')b_\alpha(E''')\rangle - \langle a_\alpha^\dagger(E'')a_\alpha(E''')\rangle) \Bigg].$$

We consider the charge-pumped fluctuations, which are defined as

$$\delta Q_{\alpha\alpha} = \int_0^{T_0} dT \int_{-\infty}^\infty d\tau\, \delta\left(T + \frac{\tau}{2}, T - \frac{\tau}{2}\right). \tag{E.8}$$

In the latter, we employ the Wick theorem and cancel the disconnected averages, then insert Eq. E.3. The expression is further simplified by employing the representation of the delta function $\delta(\alpha) = \int_{-\infty}^\infty dt\, e^{i\alpha t}$. We obtain all the expressions of [18]

$$\delta Q_{\alpha\alpha} = \delta Q_{\alpha\alpha}^{th} + \delta Q_{\alpha\alpha}^{sh}, \tag{E.9}$$

$$\delta Q_{\alpha\alpha}^{th} = 2T_0 \int \frac{d\epsilon}{2\pi} f(\epsilon)\tilde{f}(\epsilon) \sum_n (1 - |S_{\alpha\alpha}^F(\epsilon_n, \epsilon)|^2), \tag{E.10}$$

and

$$\delta Q_{\alpha\alpha}^{sh} = T_0 \int \frac{d\epsilon}{(2\pi)^2} \sum_{\gamma\delta} \sum_n \sum_m \sum_p \frac{(f(\epsilon_n) - f(\epsilon_m))^2}{2} (S_{\alpha\gamma}^{F*}(\epsilon,\epsilon_n) S_{\alpha\delta}^F(\epsilon,\epsilon_m)$$

$$\times S_{\alpha\delta}^{F*}(\epsilon_p,\epsilon_m) S_{\alpha\gamma}^F(\epsilon_p,\epsilon_n)). \tag{E.11}$$

The Floquet scattering matrix has the following adiabatic expansion

$$S^F(E_n, E) = S_n(E) + \frac{n\Omega}{2}\frac{\partial}{\partial\epsilon}S_n(E) + \Omega A_n + O(\Omega^2), \tag{E.12}$$

with $S_n(E)$ the n-th Fourier coefficient of the scattering matrix defined as

$$S_n(E) = \int_0^{T_0} \frac{dt}{T_0} e^{in\Omega t} S(E), \tag{E.13}$$

and $A_n$ is the first order correction of the quantity. By substituting the expansion, one obtains the zero and first-order thermal noise

$$\delta Q_{\alpha\alpha}^{0,th} = 2k_B T \int_{-\infty}^\infty \frac{d\epsilon}{2\pi}\left(-\frac{df}{d\epsilon}\right)\left[T_0 - \int_0^{T_0} dT |S_{\alpha\alpha}(E)|^2\right], \tag{E.14}$$

and

$$\delta Q_{\alpha\alpha}^{1,th} = k_B T \int_{-\infty}^\infty \frac{d\epsilon}{4\pi i} \int_0^{T_0} dT\left(-\frac{df}{d\epsilon}\right) \sum_{\beta\neq\alpha} \frac{dI_{\alpha\alpha}}{dE}, \tag{E.15}$$

where $\frac{dI_{\alpha\alpha}}{dE}$ is the spectrally resolved current, with its definition

$$\frac{dI_{\alpha\alpha}}{dE} = \left(\frac{\partial S_{\alpha\alpha}^*}{\partial t}S_{\alpha\alpha} - \frac{\partial S_{\alpha\alpha}}{\partial t}S_{\alpha\alpha}^*\right). \tag{E.16}$$

These two expressions can correspond with the ones obtained from the gradient expansion.

The shot noise term has different expressions according to the regime in which we consider it. In the zero temperature limit $k_B T \ll \Omega$ and $k_B T \ll \Gamma^{-1}$. In the zero temperature limit, the difference $(f(E_n) - f(E_m))^2 \simeq \theta(E_m - \mu) - \theta(E_n - \mu)$. Taking the other scattering matrix elements in the zero order

$$\delta Q_{\alpha\alpha}^{1,sh} = \sum_{q=1}^{\infty} \frac{q\Omega}{8\pi^2} C_{\alpha\alpha}^{(sym)}(\mu). \tag{E.17}$$

In the high-temperature limit, instead, we can formally write the difference as $(f(E_n) - f(E_m))^2 \simeq \left(-\frac{df}{d\epsilon}\right)|n - m|\Omega$. Then this "high-temperature shot noise" reads

$$\delta Q_{\alpha\alpha}^{2,sh} = \int_{-\infty}^{\infty} \frac{d\epsilon}{8\pi^2} \left(\frac{df}{d\epsilon}\right)^2 \sum_{q=1}^{\infty} (q\Omega)^2 C_{\alpha\alpha}^{(sym)}(E). \tag{E.18}$$

Note that this belongs to the second order in the adiabatic expansion. This expression can be understood by rewriting it in terms of the derivatives and compared with the gradient expansion. In fact, we have that

$$\sum_q q\Omega [A]_q = \frac{1}{i} \partial_t A(\epsilon). \tag{E.19}$$

Then

$$\delta Q_{\alpha\alpha}^{(2,sh)} = -\int_0^{T_0} dT \int \frac{d\epsilon}{16\pi} (\partial_\epsilon f(\epsilon))^2 \left\{ 2\sum_\beta \left[ \partial_T^2 S_{\alpha\beta} S_{\alpha\beta}^\dagger + S_{\alpha\beta} \partial_T^2 S_{\alpha\beta}^\dagger - 2\partial_T S_{\alpha\beta} \partial_T S_{\alpha\beta}^\dagger \right] \right.$$
$$\left. - 2\sum_{\gamma\delta} (\partial_T S_{\alpha\delta} S_{\alpha\delta}^\dagger - S_{\alpha\delta} \partial_T S_{\alpha\delta}^\dagger)(\partial_T S_{\alpha\gamma} S_{\alpha\gamma}^\dagger - S_{\alpha\gamma} \partial_T S_{\alpha\gamma}^\dagger) \right\}. \tag{E.20}$$

The relevance of all these terms is discussed in the next section of the Appendix.

Using this formalism of the Floquet scattering matrix, now we show how to derive the expansion of all the quantities we have analyzed in the main article. In terms of the operators, the current reads [13]

$$I_\alpha(t) = \int \frac{dE dE'}{(2\pi)^2} e^{i(E-E')t} (b_\alpha^\dagger(E) b_\alpha(E') - a_\alpha^\dagger(E) a_\alpha(E')). \tag{E.21}$$

The charge pumped is the integral over t of this quantity. After substituting the definition of the Floquet scattering matrix, we have the following expression

$$Q_\alpha = T_0 \int \frac{dE}{2\pi} \sum_\beta \sum_n (|S_{\alpha\beta}^F(E, E_n)|^2 f(E_n) - f(E)). \tag{E.22}$$

When shifting the energy variables $E \to E - n\Omega$

$$Q_\alpha^{(1)} = T_0 \int \frac{dE}{2\pi} \sum_\beta \sum_n |S_{\alpha\beta}^F(E_n, E)|^2 (f(E) - f(E_n)). \tag{E.23}$$

The difference of Fermi functions $f(E) - f(E_n) \to n\Omega\left(-\frac{df}{d\epsilon}\right)$, formally assuming $k_B T \gg \Omega$. However, one can demonstrate that this gives the correct result in the zero temperature limit as well. Inserting eq. E.12, one obtains the following expression

$$Q_\alpha^{(1)} = T_0 \int \frac{dE}{2\pi} \sum_\beta \sum_n \left(-\frac{df}{d\epsilon}\right) n\Omega |S_{\alpha\beta}^n(E)|^2. \tag{E.24}$$

Using relation E.19, we obtain our previous expression of the pumped charge

$$Q_\alpha^{(1)} = \int_0^{T_0} dt \int \frac{dE}{2\pi} \sum_\beta \left( \frac{\partial S_{\alpha\beta}^*}{\partial t} S_{\alpha\beta} - S_{\alpha\beta}^* \frac{\partial S_{\alpha\beta}}{\partial t} \right). \tag{E.25}$$

The variation of the number of particles is obtained from the currents as $\dot{N}^{(i)} = \sum_\alpha I_\alpha^{(i)}$

$$\dot{N}^{(1)} = \sum_\alpha \left[ \dot{\epsilon}_d \int \frac{d\epsilon}{4\pi i} \frac{2i\Gamma_\alpha A}{\Gamma} + \dot{\Gamma}_\alpha \int \frac{d\epsilon}{4\pi i} 2i Re G^R \right] = \dot{\epsilon}_d \int \frac{d\epsilon}{2\pi} A + \dot{\Gamma} \int \frac{d\epsilon}{2\pi} Re G^R. \tag{E.26}$$

Likewise for the heat current

$$I_{H,\alpha} = \int_{-\infty}^\infty \frac{dE}{2\pi} \sum_n (E_n - \mu) \sum_\beta |S_{\alpha\beta}^F(E_n, E)|^2 (f(E) - f(E_n)). \tag{E.27}$$

Employing the same reasoning as before, we write the first-order heat exchange as

$$I_{H,\alpha}^{(1)} = \int_{-\infty}^\infty \frac{dE}{4\pi i} (E - \mu) \left( \frac{\partial S_{\alpha\beta}^*}{\partial t} S_{\alpha\beta} - S_{\alpha\beta}^* \frac{\partial S_{\alpha\beta}}{\partial t} \right), \tag{E.28}$$

and

$$\mathcal{Q}^{(1)} = \sum_\alpha I_{H,\alpha}^{(1)}. \tag{E.29}$$

The energy current reads

$$I_{E,\alpha} = \int_{-\infty}^\infty \frac{dE}{2\pi} E \sum_\beta \sum_n |S_{\alpha\beta}^F(E_n, E)|^2 (f(E) - f(E_n)), \tag{E.30}$$

and at first order

$$\dot{E}^{(1)} = \dot{\epsilon}_d \int_{-\infty}^\infty \frac{dE}{2\pi} E (-\partial_\epsilon f) A + \dot{\Gamma} \int_{-\infty}^\infty \frac{dE}{2\pi} E (-\partial_\epsilon f) Re G^R. \tag{E.31}$$

Finally, we introduce the entropy current with lead $\alpha$, which has the following expression [32]

$$I^\Sigma = k_B \int \frac{dE dE'}{(2\pi)^2} e^{i(E-E')t} \left[ \log f (b_\alpha^\dagger(E) b_\alpha(E') + a_\alpha^\dagger(E) a_\alpha(E')) + \log(1-f)(b_\alpha(E) b_\alpha^\dagger(E') \right.$$
$$\left. + a_\alpha(E) a_\alpha^\dagger(E')) \right]. \tag{E.32}$$

In terms of the Floquet scattering matrix, the latter reads

$$I^\Sigma = \int \frac{dE}{2\pi} \frac{(E-\mu)}{T} \sum_n (E_n - \mu) \sum_\beta |S_{\alpha\beta}^F(E_n, E)|^2 (f(E) - f(E_n)). \tag{E.33}$$

Repeating the analysis of the current, we end up with this expression

$$\dot{S}^{(1)} = \dot{\epsilon}_d \int_{-\infty}^\infty \frac{dE}{2\pi} \frac{(E-\mu)}{T} (-\partial_\epsilon f) A + \dot{\Gamma} \int_{-\infty}^\infty \frac{dE}{2\pi} \frac{(E-\mu)}{T} (-\partial_\epsilon f) Re G^R. \tag{E.34}$$

The work rate $\dot{W}$ can be obtained by using the first law of thermodynamics. This analysis can be extended to further orders and concludes that all our results coincide with the gradient expansion method.

## F Comparison between the leading order terms of noise in the various regimes

As pointed out in [18], there are three different regimes in which the different leading order terms of the noise are relevant. We have three relevant terms for our analysis. The first is the first-order thermal noise arising from the thermal excitation of the scatterer, corresponding to Eq. 25. It contains an energy scale proportional to $k_B T \frac{\Omega T_0}{\Gamma}$. The second term is the shot noise term (Eq. 26), which contains an energy scale of $\Omega T_0$. The third term is the second-order adiabatic shot noise in the high-temperature limit of Eq. E.18 This term is, strictly speaking, singular in the zero-temperature limit and it contains an energy scale of $\frac{\Omega^2 T_0}{k_B T}$.

By examining the ratio of these terms, we determine the regimes in which each of these terms is relevant. We can conclude that in the low-temperature regime $K_B T \ll \Omega$, the first-order "shot" noise is prevalent. There is an intermediate temperature regime $\Omega \ll k_B T \ll \sqrt{\Omega \Gamma}$ in which the high-temperature "shot" noise is prevalent, while in the high-temperature limit $\sqrt{\Omega \Gamma} \ll k_B T$.

## G The adiabaticity conditions

In this section, we analyze the conditions on the instantaneous velocity and acceleration according to which the physical process we are considering can be adiabatic. Following the method put forward by [33], we extend the adiabatic expansion of the charge pumped up to the third order and require that the first order in the expansion of the charge current is much bigger than the higher order corrections. The first-order instantaneous current can be rewritten in the form

$$Q_\alpha^{(1)} = \frac{1}{T_0} \int_0^{T_0} dt \sum_i A_{\alpha,i}(t) \frac{dx_i}{dt} \,. \tag{G.1}$$

This is the adiabatic term, which can be interpreted in a geometrical manner (Brouwer's formula). The second order has two contributions

$$Q_\alpha^{(2)} = \frac{1}{T_0} \int_0^{T_0} dt \left( \sum_i B_{\alpha,i}^{(1)}(t) \frac{d^2 x_i}{dt^2} + \sum_{i,j} B_{\alpha,i,j}^{(2)}(t) \frac{dx_i}{dt} \frac{dx_j}{dt} \right). \tag{G.2}$$

Using integration by parts, the second order correction becomes

$$B_{\alpha,i,j} = B_{\alpha,i,j}^{(2)} - \frac{\partial B_{\alpha,i}^{(1)}(t)}{\partial x_j} \,. \tag{G.3}$$

Exactly in the same way, from the third order we can distinguish 2 different terms:

$$\langle I_\alpha^{(3)}(t) \rangle = \langle I_\alpha^{(3v)}(t) \rangle + \langle I_\alpha^{(3a)}(t) \rangle \,, \tag{G.4}$$

$$\langle I_\alpha^{(3v)}(t) \rangle = \sum_{i,j,k} C_{\alpha,i,j,k}(t) \frac{dx_i}{dt} \frac{dx_j}{dt} \frac{dx_k}{dt} \,, \tag{G.5}$$

$$\langle I_\alpha^{(3a)}(t) \rangle = \sum_{i,j} D_{\alpha,i,j}(t) \frac{d^2 x_i}{dt^2} \frac{dx_j}{dt} \,. \tag{G.6}$$

In order for the process to be adiabatic, we must require that the first order in the expansion of the charge current is much bigger than the higher order corrections:

$$|\langle I_\alpha^{(1)}(t) \rangle| \gg |\langle I_\alpha^{(2)}(t) \rangle|, |\langle I_\alpha^{(3v)}(t) \rangle|, |\langle I_\alpha^{(3a)}(t) \rangle| \,. \tag{G.7}$$

We translate this condition in terms of the coefficients. In this way, we can rewrite the adiabaticity condition for the second-order correction defining a velocity limit $v_{lim,\alpha}^{(2)}(t)$ for which it must be true that

$$|v(t)| \ll v_{lim,\alpha}^{(2)}(t). \tag{G.8}$$

The velocity limit can be defined as

$$v_{lim,\alpha}^{(2)}(t) = \frac{|A_\alpha(t)|}{|B_\alpha(t)|}, \tag{G.9}$$

$$A_\alpha(t) = \sum_i A_{\alpha,i}(t)\tilde{v}_i(t), \tag{G.10}$$

$$B_\alpha(t) = \sum_{i,j} B_{\alpha,i,j}(t)\tilde{v}_i(t)\tilde{v}_j(t), \tag{G.11}$$

where $\tilde{v}_i = \frac{v_i}{|v_i|}$. For the third order correction in similar way

$$v_{lim,\alpha}^{(3)}(t) = \sqrt{\frac{|A_\alpha(t)|}{|C_\alpha(t)|}}, \tag{G.12}$$

$$C_\alpha(t) = \sum_{i,j,k} C_{\alpha,i,j,k}(t)\tilde{v}_i(t)\tilde{v}_j(t)\tilde{v}_k(t). \tag{G.13}$$

So, in the end we must require that

$$v(t) \ll \min[v_{lim}^{(2)}, v_{lim}^{(3)}, ...]. \tag{G.14}$$

Now, let's examine these conditions on our peristaltic cycle, for which $x_1 = \epsilon_d$ and $x_2 = \delta\Gamma$.

We can extract an energy scale $\frac{1}{\Gamma^2}$ and express in terms of the adimensional variables $x = \epsilon_d/\Gamma$ and $y_0 = \delta_\Gamma/\Gamma$

$$|A_{\alpha,1}| = \frac{2}{\pi\Gamma}\frac{1 \pm y_0}{2}\left|\frac{1}{x^2+1} + \frac{2x^2}{(1+x^2)^2} - \frac{2}{(1+x^2)^2}\right|. \tag{G.15}$$

In the same way, we can compute $A_{\alpha,2}$. The result is

$$|A_{\alpha,2}| = \frac{1}{4\pi\Gamma}\left|\frac{x_0}{1+x_0^2}\right|, \tag{G.16}$$

in terms of $x_0 = \epsilon_0/\Gamma$. The results of the second-order coefficients are

$$B_{\alpha,11}(t) = \frac{2}{\pi\Gamma^3}\frac{1 \pm y_0}{2}\partial_\epsilon A^2. \tag{G.17}$$

The resulting condition for the derivative of $\epsilon_d$ is

$$\frac{\dot{\epsilon_d}}{\Gamma^2} \ll f_2(x, y_0), \tag{G.18}$$

where $f_2$ is an adimensional function which is in fig.15.

Now, let's analyze the third-order corrections for the velocity. One can obtain the expression of the coefficient

$$C_{\alpha,111} = \frac{1}{2\pi\Gamma^5}\frac{1 \pm y_0}{2}\left\{\left(\frac{1}{(x-i)^6} + \frac{1}{(x+i)^6}\right) + i\frac{2}{1+x^2}\left(-\frac{1}{(x-i)^5} + \frac{1}{(x+i)^5}\right)\right.$$
$$\left. -i\frac{2}{(1+x^2)^2}\left(-\frac{1}{(x-i)^3} + \frac{1}{(x+i)^3}\right) - \frac{8}{(1+x^2)^4}\right\}. \tag{G.19}$$

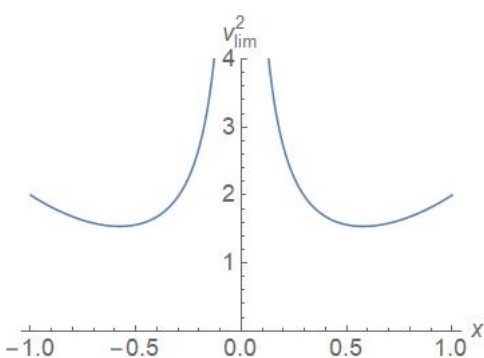

Figure 15: The adimensional function $f_2(x, y_0)$ with respect to the variable x, for $y_0 = 1$.

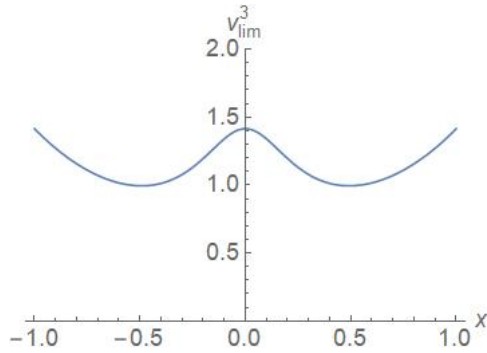

Figure 16: The adimensional function $f_3(x, y_0)$ with respect to the variable x, for $y_0 = 1$.

The coefficient $C_{\alpha,222}$ is also equal to 0. Therefore, we can express the condition $v(t) \ll v_{lim}^{(3)}$ as

$$\frac{\dot{\epsilon}_d}{\Gamma^2} \ll f_3(x, y_0), \tag{G.20}$$

where $f_3(x, y_0)$ is in 16.

Finally, let's consider the limits on acceleration. From the adiabatic expansion, we can infer that the coefficient $D_{\alpha11}$ is

$$D_{\alpha11} = \frac{1}{3\pi} \frac{1 \pm y0}{2} \left\{ \left( \frac{1}{(x-i)^5} + \frac{1}{(x+i)^5} \right) - i\frac{2}{1+x^2} \left( \frac{1}{(x+i)^4} - \frac{1}{(x-i)^4} \right) \right.$$
$$\left. - i\frac{34}{(1+x^2)^2} \left( \frac{1}{(x+i)^2} - \frac{1}{(x-i)^2} \right) \right\}. \tag{G.21}$$

The resulting condition on the acceleration is

$$\frac{\ddot{\epsilon}_d}{\Gamma^3} \ll g(x, y_0), \tag{G.22}$$

$g(x, y_0)$ is in (fig.17). These results appear to justify the claim that the adiabatic expansion is well-defined along the entire cycle, provided that the appropriate bounds on the derivatives of the time-dependent quantities are respected. However, repeating the same reasoning with the current noise and the thermodynamic rates would signal that there are divergences when one of the couplings with the two baths is switched off: $\Gamma_L = 0$ or $\Gamma_R = 0$.

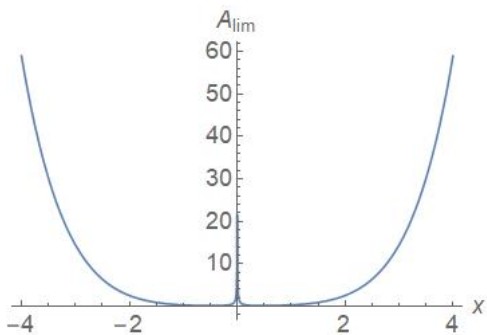

Figure 17: The adimensional function $g(x, y_0)$ with respect to the variable x, for $y_0 = 1$.

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
