# Peer review of "Thermodynamics of adiabatic quantum pumping in quantum dots"

_SciPost Physics Core, doi:SciPost Phys. Core 7, 067 (2024)_

## Round 1 · Referee Report · Anonymous (Referee 2) · 2024-5-24

Report

The authors have revised the manuscript and addressed the questions raised. The present version is suitable for the publication in SciPost Physics core.

Recommendation

Publish (meets expectations and criteria for this Journal)

  • validity: -
  • significance: -
  • originality: -
  • clarity: -
  • formatting: -
  • grammar: -

Author:  Daniele Nello  on 2024-09-12  [id 4771]

(in reply to Report 1 on 2024-05-24)

We thank the Reviewer for their positive feedback on the article.

---

## Round 1 · Referee Report · Anonymous (Referee 1) · 2024-6-2

Report

In the revised version the authors properly addressed my comments. I particularly appreciate the clarifications about the averages and about the noise in the triangular cycle. At the same time, the reorganization of the paper has also done some damage. In several places some quantities now appear without being defined. Here are the specific points related to this and other issues.

  1. The leads L and R are sometimes labeled by the index i, sometimes by the index alpha. Could the authors please choose one and use it everywhere? (I would prefer alpha.)

  2. In Eq. (10) f is not defined at all, and S is defined only by words which is a bit ambiguous. This happened because the derivation of Eq.(10) was moved to Appendix A (which was probably a good idea in general). But there is no reference to Appendix A around Eq. (10) in the revised version. Citing Appendix A here would be helpful.

  3. After Eq.(24), it is beta that is inverse temperature, rather than 1/beta, isn't it?

  4. Just before Eq.(26), Omega is not defined. My brave guess would be that Omega = 2 pi / T_0 and not the grand-canonical potential introduced a few paragraphs later. Please define.

  5. Eq.(29) is a complete disaster: the integration is over t, in the exponential there is Omega and no t, and A has no argument at all.

  6. After Eq.(63), delta appears without definition. In the original version, it was mentioned that introducing delta was necessary in order to avoid divergences due to the breakdown of adiabatic approximation. That was a very useful remark, but for some reason it disappeared from the revised version. May the authors please put it back?

  7. In Sec. 5.1 deltaGamma is not defined, neither in Fig.1, nor in the text. We can only guess that Gamma_L = (Gamma + deltaGamma) / 2 and Gamma_R = (Gamma - deltaGamma) / 2 during the peristaltic cycle. And in Fig.1 the barely readable and not very useful numbers go in the order 1-2-4-3 - what is this supposed to mean? The initial and final positions of the dot level are sometimes called epsilon_1 and epsilon_2, and sometimes +epsilon_0 and -epsilon_0. In addition, epsilon also denotes the ratio delta/Gamma. I beg the authors to clean up all this mess.

  8. Figures 2 and 7 would better be plotted in log scale, otherwise most of the plot (with x_0 > 20) is not very informative.

  9. I don't understand how is Eq.(70) "signalling the presence of a quantization rule for the work rate". I don't see any quantization here. This expression is valid only for constant level velocity; if the level position changed by some other law, the work would probably be proportional to some typical value of the level velocity, but there is no natural quantity which would play the role of the work quantum. The work per cycle is just finite, that's all I can see here.

The process seems to be slowly converging towards publication in SciPost Physics Core. The current version of the manuscript still looks too much like working notes of a student who enjoys doing calculations but hates making figures (with all my appreciation of the hard work that was done here). It would be good if the supervisor takes a look at the next version before the resubmission.

Recommendation

Ask for major revision

  • validity: -
  • significance: -
  • originality: -
  • clarity: -
  • formatting: -
  • grammar: -

Author:  Daniele Nello  on 2024-09-12  [id 4773]

(in reply to Report 2 on 2024-06-02)
Category:
remark
answer to question
reply to objection
correction

We would like to thank the Reviewer for their useful comments. In the revised version, we have sought to address all the points raised in the Report. In particular:
**Comment 1 ** : "The leads L and R are sometimes labelled by the index i, sometimes by the index alpha. Could the authors please choose one and use it everywhere? (I would prefer alpha.)"
**Reply** : The labelling of the leads has been changed to $\alpha=L,R$ along all the article.
**Comment 2** :" In Eq. (10) f is not defined at all, and S is defined only by words which is a bit ambiguous. This happened because the derivation of Eq.(10) was moved to Appendix A (which was probably a good idea in general). But there is no reference to Appendix A around Eq. (10) in the revised version. Citing Appendix A here would be helpful."
**Reply**: The definition of the Fermi function has been included after Eq. (10) of the current version and a reference to Appendix A has been added.
**Comment 3**:"After Eq.(24), it is beta that is inverse temperature, rather than 1/beta, isn't it?"
**Reply**: The mistake has been corrected in the new version.
**Comment 4**: " Just before Eq.(26), Omega is not defined. My brave guess would be that $\Omega = 2 \pi / T_0$ and not the grand-canonical potential introduced a few paragraphs later. Please define."
**Reply**: The definition has been included in the new version.
**Comment 5**:"Eq.(29) is a complete disaster: the integration is over t, in the exponential there is Omega and no t, and A has no argument at all."
**Reply**: In Eq. (29) there is a missing t in the exponential. Of course, A depends on t and this has been pointed out in the new version.
**Comment 6**:" After Eq.(63), delta appears without definition. In the original version, it was mentioned that introducing delta was necessary in order to avoid divergences due to the breakdown of adiabatic approximation. That was a very useful remark, but for some reason it disappeared from the revised version. May the authors please put it back?"
**Reply**: A definition of $\delta\Gamma$ has been included in Sec. 5.1, coinciding with the Reviewer's guess. The remark about the breakdown of the adiabatic approximation has been put back where it was and the normalized regularization parameter has been renamed to $\xi=\delta/\Gamma$.
**Comment 7**:"In Sec. 5.1 deltaGamma is not defined, neither in Fig.1, nor in the text. We can only guess that $\Gamma_L = (\Gamma + \delta\Gamma) / 2$ and $\Gamma_R = (\Gamma - \delta\Gamma) / 2$ during the peristaltic cycle. And in Fig.1 the barely readable and not very useful numbers go in the order 1-2-4-3 - what is this supposed to mean? The initial and final positions of the dot level are sometimes called $\epsilon_1$ and $\epsilon_2$, and sometimes $+\epsilon_0$ and $-\epsilon_0$. In addition, epsilon also denotes the ratio delta/Gamma. I beg the authors to clean up all this mess."
**Reply**: The numbers in Fig. 1 have been put in clockwise ascending order and the endpoints in the $\epsilon_d$ axis have been set to $-\epsilon_0$ and $\epsilon_0$.
**Comment 8**: " Figures 2 and 7 would better be plotted in log scale, otherwise most of the plot (with $x_0 > 20$) is not very informative."
**Reply**: The suggestion of the Reviewer has been accepted and the log scale has been introduced in Figg. 2 and 7.
**Comment 9**:" I don't understand how is Eq.(70) "signalling the presence of a quantization rule for the work rate". I don't see any quantization here. This expression is valid only for constant level velocity; if the level position changed by some other law, the work would probably be proportional to some typical value of the level velocity, but there is no natural quantity which would play the role of the work quantum. The work per cycle is just finite, that's all I can see here."
**Reply**: We agree that this concept can be confusing in this framework. We signal that the results for the work indicate that this quantity is finite and proportional to the charge pumped if we investigate the case of constant velocity.

---

## Round 1 · Referee Report · Anonymous (Referee 3) · 2024-6-7

Report

The authors properly revised the manuscript and addressed the issues I raised in my first report as well as those raised by the other referee. I believe the paper is very interesting and clearly written. The present version is suitable for publication in SciPost Physics.

Recommendation

Publish (easily meets expectations and criteria for this Journal; among top 50%)

  • validity: -
  • significance: -
  • originality: -
  • clarity: -
  • formatting: -
  • grammar: -

Author:  Daniele Nello  on 2024-09-12  [id 4772]

(in reply to Report 3 on 2024-06-07)

The authors would like to thank the Reviewer for their appreciation of our article.

---

## Editorial Decision

published